# Learning to Be Fair: Modeling Fairness Dynamics by Simulating Moral-Based Multi-Agent Resource Allocation

**Haiyan Feng**[1*]**, Yuqiao Du**[1*]**, Huacong Tang**[2]**, Junjie Liao**[3]**, Yipeng Kang**[4]**,**
**Mingjie Bi**[4✉]**, Fangwei Zhong**[3,4✉]**, Zhou Ziheng**[2✉]

[1]Tsinghua University, Beijing, China    [2]University of California, Los Angeles, CA, USA
[3]Beijing Normal University, Beijing, China
[4]State Key Laboratory of General Artificial Intelligence, Beijing Institute for General Artificial Intelligence (BIGAI), Beijing, China
[*]Equal contribution.    ✉Corresponding authors:
fangweizhong@bnu.edu.cn, bimingjie@bigai.ai, josephziheng@ucla.edu

## Abstract

Fairness is a foundational social construct for stable, resilient societies, yet its meaning is dynamic, context-dependent, and inherently subjective. This multi-faceted nature reveals a gap between traditional social science and contemporary computational approaches: the former offers rich conceptual accounts but limited computational models, while the latter often relies on static objectives or purely data-driven criteria that overlook the subjective and communicative nature of fairness. We address this gap through a computational framework and two resource-allocation scenarios in which large language model (LLM)–based cognitive agents operate with heterogeneous roles, relationships, and moral commitments. The framework supports agent reflection and negotiation via explicit, language-based feedback, enabling the study of norm evolution and consensus formation of fairness in multi-agent social systems. Using standard objective metrics from resource allocation, we analyze how our approach captures the complexities of fairness, such as ambiguity, procedural justice, and subjective satisfaction—while remaining quantitatively evaluable. This work also offers insights for designing tractable AI systems that can navigate evolving social norms in dynamic, multi-stakeholder environments.

## 1 Introduction

Fairness is a cornerstone of human decision-making and social stability, yet defining what is "fair" is inherently complex. Unlike a fixed axiom, fairness is better understood as a socially constructed and context-dependent concept (Deck et al., 2024). Human judgments of fairness can shift dramatically with changing roles, situations, and cultural norms, reflecting the dynamic nature of fairness evaluations. Moreover, perceived fairness is a subjective attitude strongly tied to context and moral reasoning. Individuals often disagree on fairness due to divergent personalities and preferences, which shape what different people consider fair, adding another layer of complexity (Kurschilgen, 2023; Dai & Xiao, 2025). This intrinsic complexity of fairness has been linked to real societal implications like institutional trust and societal well-being (Ma et al., 2024). Therefore, fairness is important because it is an evolving consensus of values. Understanding how fairness is perceived and negotiated across contexts is essential for tackling societal issues (inequality, trust in governance, group conflicts), especially as AI systems increasingly mediate these high-stakes decisions.

Social scientists have noted that norms of fairness can emerge or change as a group interacts repeatedly, learning what is acceptable or not (Young, 1993). These evolutionary dynamics of fairness are inextricably linked to human decision-making: individuals' sense of fairness influences their decisions, while the decisions can reshape their fairness norms over time. A typical scenario to study these dynamics is resource allocation, where multiple parties must agree on how to distribute lim-

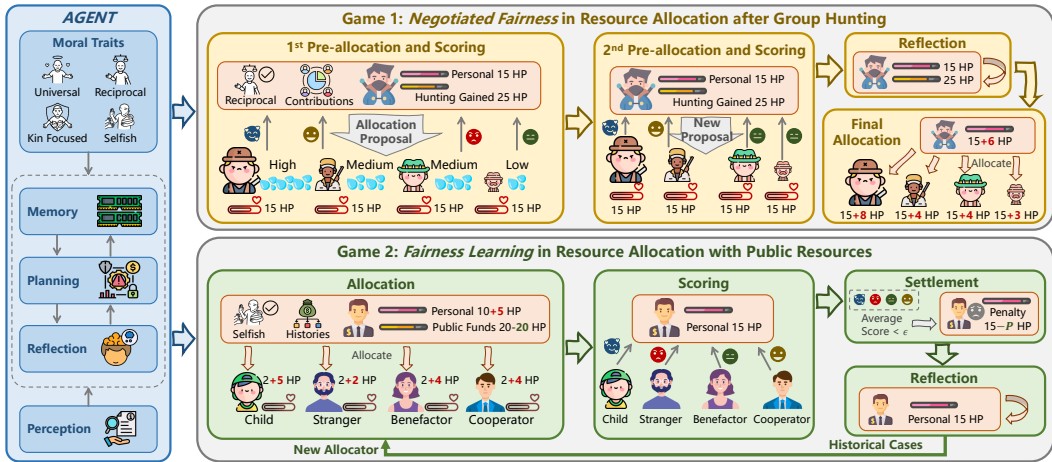

Figure 1: Our proposed agent architecture and simulation mechanisms for two resource allocation games: Negotiated Fairness and Fairness Learning.

ited resources (Moulin, 2004). For instance, in repeated bargaining, individuals often adjust their concept of a fair split based on previous outcomes and the behavior of others (Slembeck, 1999). Punishment of norm violators sustains cooperation even when costly, knitting expectations of "what is fair" into behavior over time (Fehr & Gächter, 2002). In this paper, we focus on two key research questions arising from this perspective: 1) how do agents' reflection and social interactions influence the evolution and convergence (or divergence) of a group fairness consensus? 2) how do differences in scenario context or moral types lead to gaps between what is "objectively" fair by some metric versus what agents subjectively accept as fair?

Traditional research on fairness in the fields of economics, psychology, and political science provide rich insights, conceptual descriptions, and models of fairness, such as inequality aversion (Fehr & Schmidt, 1999), equity/reciprocity (Bolton & Ockenfels, 2000), group (e.g., demographic parity (Hardt et al., 2016), etc. These studies have limited ability to address our focused problems, but as AI techniques develop, multi-agent systems have been paid attention to investigate fairness, specifically using multi-agent reinforcement learning (MARL) and LLM-based simulation. However, MARL incorporates "fairness" primarily via reward shaping or static welfare objectives (Hughes et al., 2018; Jaques et al., 2019a), while LLM agent-based simulation enables complex fairness representation but lacks a feedback mechanism (Park et al., 2023; Gao et al., 2024). Consequently, a gap remains for a computational framework that captures the complexity of fairness and integrates subjective feedback/rationales to study fairness evolution and consensus under moral heterogeneity.

To address these limitations, we introduce an LLM-based multi-agent simulation framework to study fairness as an evolving construct. Our contributions include: 1) a reflection and feedback mechanism in our LLM-based simulated environment where cognitive agents treat fairness as a context-sensitive and evolving construct; 2) two simulation mechanisms (Negotiated Fairness and Fairness Learning) to investigate the evolving dynamics of resource allocation rules and fairness consensus; 3) a comparison between objective fairness metrics with human-like subjective satisfaction, analyzing their alignment or divergence; 4) a controlled simulation testbed for examining how agents with reflection and language-based reasoning capture ambiguity, procedural justice, and moral heterogeneity.

**Scope and intended use.** Our goal is not to propose a deployable fair-allocation algorithm or to claim that simulated allocations should be used as prescriptive guidance in real-world settings. Instead, we use controlled multi-agent simulations to study a narrower question: how repeated social feedback, moral heterogeneity, and scenario context shape the relationship between objective allocation criteria and socially accepted allocations. We therefore position the framework as a hypothesis-generating testbed for fairness dynamics, rather than a normative decision system.

## 2 RELATED WORK

**Classic Fairness Theories** Traditional research on fairness emphasizes its multidimensional nature and centrality to legitimacy. They distinguish distributive, procedural, and interactional fairness, highlighting that people value both outcomes and processes, reasons, and dignified treatment (Rawls, 1971; Tyler, 1990; Bies & Moag, 1986). They provide a rich but descriptive conceptual foundation, lacking computational models of how fairness judgments evolve with repeated interaction.

**Computational Fairness Models** Economists and computational social scientists formalize fairness through fair-division rules (e.g., envy-freeness, Nash welfare; (Caragiannis et al., 2016; Budish, 2011)) and outcome-based behavioral models of inequity aversion and reciprocity (Fehr & Schmidt, 1999; Bolton & Ockenfels, 2000). These approaches render fairness as operational indices but rely on fixed formulas or equilibrium assumptions, emphasizing end-state allocations over the dynamic, negotiated, and subjective character of fairness. Machine learning extends this trajectory by embedding fairness into algorithms via parity constraints, causal adjustments, or contestability mechanisms (Hardt et al., 2016; Dwork et al., 2012; Feldman et al., 2015). Although these models improve on rigid formulas through data-driven generalization, they remain largely one-shot and model-centric, treating fairness as an external constraint rather than an evolving consensus.

**Multi-Agent Simulations** As AI advances, researchers turn to agent-based simulations in which fairness emerges from interaction. In MARL, cooperation and fairness can be sustained when social preferences are embedded in rewards: inequity aversion improves cooperation in intertemporal dilemmas (Hughes et al., 2018), prosocial utility shifts equilibria toward fairer outcomes (Peysakhovich & Lerer, 2017), intrinsic social-influence rewards foster emergent communication (Jaques et al., 2019b), and fairness on reward sharing (Huang et al., 2024). These findings underscore MARL's strength—policy adaptation through feedback—yet fairness remains pre-specified in the reward rather than negotiated. By contrast, large language model (LLM) agents excel at rich reasoning, justification, and natural negotiation (Gao et al., 2024). In multi-agent collaboration, they exhibit emergent theory-of-mind and cooperative behavior (Li et al., 2023b), while generative-agent simulations show how language-driven memory and reflection yield believable collective dynamics (Park et al., 2023). Recent frameworks extend this line: AutoGen offers a conversation-based coordination architecture (Wu et al., 2023), CAMEL explores role-based communicative societies (Li et al., 2023a), Social-Evol introduces LLM cognitive agents with morality (Ziheng et al., 2025), and systematic evaluations highlight both promise and limits of LLM negotiators (Kwon et al., 2024; Abdelnabi et al., 2024). Studies of consensus seeking also reveal biases toward simplistic averaging (Chen et al., 2023). These works demonstrate the potential of language-native interaction to capture subjective, procedural aspects of fairness; however, because LLM policies are fixed, agents lack a closed feedback learning loop. Despite discussing and justify fairness, they cannot adapt behavior to external signals as MARL can.

In summary, prior work on fairness either provides feedback-driven learning without justifications (MARL) or justifications without adaptive learning (LLM agents), leaving open the challenge of modeling fairness as a dynamic consensus that integrates both. Our framework directly addresses the gap by enabling cognitive LLM agents to engage in negotiation and reflection while incorporating structured feedback, thereby linking subjective moral reasoning with objective allocation metrics.

## 3 METHODOLOGY

### 3.1 LLM COGNITIVE AGENTS SIMULATION ENVIRONMENT

We developed a text-based multi-agent society with social interactions to simulate cognitive agent behaviors in resource allocation scenarios with feedback and reflection. Heterogeneous LLM-driven agents negotiate for resource allocation, receiving endogenous and exogenous feedback (e.g., peer evaluations, institutional signals), which prompts reflection and policy updates. Figure .1 shows our agent framework and game design to study how agents learn fairness and reach consensus.

**In-Context Behavioral Adaptation Learning** In this work, "learning" specifically refers to in-context behavioral adaptation, a form of social learning that does not involve parameter updates of the agent's core decision-making framework. Agents leverage memory of past interactions (e.g., feedback from prior allocation outcomes, observations of others' responses) and iterative reflection

to adjust their future decisions dynamically. This process simulates how humans implicitly learn and adapt to social norms (e.g., fairness principles in resource distribution) through real-world interactions, rather than through explicit optimization of model parameters. Notably, this definition aligns with the goal of modeling norm dynamics: it emphasizes adaptive behavior rooted in experiential knowledge, ensuring the agent's learning process is both psychologically plausible and consistent with the study of social norm emergence.

**Environment Settings** Agents are endowed with health points (HPs) that serve both as a survival constraint and an allocatable resource. They determine how to allocate their HP across scenarios and social relationships. The environment provides two mechanisms for resource exchange and coordination: 1) *Allocation:* agents decide to transfer HP to others, enabling cooperative behavior and care-oriented ties; and 2) *Communication:* agents can transmit messages to targeted individuals or concurrently to groups, supporting coordination and information sharing.

**Simulation Process** Each simulation round is an isolated action step where agents are situated in controlled moral dilemmas, making it observable how distinct moral types and scenarios lead to concrete choices, thereby clarifying a key mechanism that underlies the broader fairness dynamics.

The system advances in discrete cycles that couple agent cognition with environmental dynamics: 1) *Environment Update* updates agent states (HP allocated and information gathered) based on actions; 2) *Agent Perception* delivers observations about the current world state and recent memories; 3) *Cognitive Processing* uses an LLM-based architecture to interpret perceptions, update memory, form judgments, and generate dispositional plans aligned with each agent's moral type toward agents; 4) *Action Planning* prioritizes among dispositional plans and codifies action plans, subject to current conditions. Each simulation step includes allocation and communication behaviors; 5) *Consequence Resolution* determines outcomes for all actions and passes them for environment updating.

**Cognitive Agent Design** Each agent is initialized with three components: 1) a profile aligned with a designated moral type (*Self-focused*, *Kin-focused*, *Reciprocal group moral*, or *Universal group moral*) (Ziheng et al., 2025); 2) a rule set specifying the environmental dynamics of the simulation; and 3) a knowledge handbook capturing common-sense assumptions about environmental processes and causal relations. Agent cognitive architecture includes: 1) *Perception Module:* Ingests the agent's internal state (e.g., HP) and recent events, consolidating them into working memory; 2) *Cognitive Processing System:* Implements an entity-centric mechanism that maintains memory, produces judgments, and forms dispositional stances toward other entities; it organizes salient information into a narrative-like situational model; 3) *Action Planning:* Translates updated memories and current constraints into concrete communication and allocation plans over the near horizon; 4) *Reflection Module:* Validates that cognition and planning are factually consistent and faithful to the agent's moral type, while ensuring outputs adhere to the required response format.

## 3.2 NEGOTIATED FAIRNESS GAME

This resource allocation game follows a group hunting event, where the team leader (allocator) divides a mixed-ownership resource pool (HPs) among four contributors with varying levels of effort. This situation creates tension between equity, efficiency, and relational fairness. The allocator balances authority and accountability, mirroring real-world leaders managing competing moral and strategic demands. To investigate how fairness consensus is dynamically co-constructed, we design a multi-turn pre-allocation and negotiation between cognitive agents, without reliance on predefined rules or historical cases. Detailed processes are described as follows:

**Simulation mechanism** Each simulation round includes six interaction processes, occurring within a single social moment, with no external historical information. The interaction processes are: 1) **1st Pre-allocation:** The allocator proposes an initial HP distribution plan including a reference strategy and specific allocations to each teammate. This *tentative claim* is to probe social reactions instead of actual resource transfer; 2) **1st Scoring:** The four teammates independently score the allocation plan, expressing their subjective evaluation. These scores provide *social feedback*, indicating the acceptability of the proposal from diverse role perspectives; 3) **2nd Pre-allocation:** The allocator, having observed the first-round scores, may revise or maintain their initial plan. This stage tests *social calibration*: whether the allocator adjusts their claim in response to feedback; 4) **2nd Scoring:** Teammates score the updated allocation plan, generating a second wave of social feedback. The evolution of scores across rounds reflects the degree of convergence in fairness perceptions; 5)

**Reflection:** The allocator performs a structured reflection on the communication process, including causal reasoning, reflection on action, and future inspiration. This cognitive step enables the allocator to synthesize feedback before final decision-making; 6) **Final Allocation:** Based on the prior negotiation and reflection, the allocator executes the definitive distribution, reflecting the outcome of the co-constructive process. This mechanism illustrates that fairness is not pre-given, but dynamically *co-constructed* through cycles of *proposal*, *evaluation*, *reflection*, and *action*, mediated by the allocator's moral reasoning and responsiveness to social cues.

**Mechanism of NonAllocators' Scoring** In this games, all nonallocator agents are assigned a Moral Type (e.g., Selfish, Kinfocused, Reciprocal) and a specific context (e.g., high contributor, low contributor). Their scoring logic uses their Moral Type and Role Perspective to evaluate the fairness of the allocation. The LLM agent first performs language-based reasoning, considering the context (e.g., their contribution, need) to decide if the allocation meets their moral expectation, and then generates a subjective satisfaction score (1–10) based on this reasoning. Recipients score based on perceived fairness, which strongly weighs their contribution and the allocation ratio they receive. A high-contributing agent will expect a proportionally higher allocation; if the allocation fails to meet this moral/equity expectation, they will give a low score, reflecting that fairness is a subjective perception despite the objective quantity.

**Evaluation Metrics** To demonstrate the dynamic of fairness, we employ two primary metrics to indicate iteration of allocation plans and subjective fairness acceptance: **1) Allocation Differences after Negotiation** This metric examines whether allocators adjust their proposals in response to negative social feedback. We compute the concession magnitude: the absolute change in total allocated resources from *1st Pre-allocation* to *2nd*, and correlate it with the average score received in *1st Scoring*. By examining the correlations, we explore the sensitivity of the allocators to the group's subjective acceptance and their level of negotiation enthusiasm. **2) Subjective Fairness** This metric assesses whether allocator agents' justifications and reflections of the fairness are consistent with their initial moral orientation. For each allocator, we analyze the justification texts in *1st* and *2nd Pre-allocation*, and the final reflection, coding for the presence of type-specific discursive patterns.

## 3.3 FAIRNESS LEARNING GAME

In this resource allocation game, an allocator with a moral trait makes HP allocation decisions in a crisis scenario, distributing public disaster relief funds and personal reserves among five recipients: themselves, their *child* (kinship), a *benefactor* (reciprocity), a *cooperator* (trust), and a *stranger* (impartiality). This setup creates moral tension between familial loyalty, debt repayment, cooperative equity, and universal fairness. The allocator acts as a semi-public resource agent who exercises control without full ownership, mirroring real-world decision-makers such as officials or leaders. To investigate how norms of fairness can emerge from individual behaviors through the transmission of social experience, we design a longitudinal simulation in which each new allocator learns from the outcomes of prior decisions, such as whether unfair allocations led to penalties, enabling the modeling of observational learning and the gradual internalization of social norms.

**Simulation mechanism** The simulation follows a multi-round, transmissive structure, with each generation of allocator completing the following four-phase cycle: **Phase 1-Allocation:** A new allocator receives its role identity and scenario context, and conducts an HP allocation to other agents. **Phase 2-Scoring:** Four recipients independently evaluate the proposal on a 1–10 scale based on their role-specific perspectives, and provide justifications for their scores. These scores reflect subjective fairness perceptions, while the justifications reveal underlying moral considerations. **Phase 3-Settlement:** The system automatically enforces a sanction rule, applying the penalty to the allocator if it gets a low score. **Phase 4-Reflection:** The allocator performs a structured reflection based on the feedback, including: 1) *Causal reasoning*: explaining why each recipient held their attitude (e.g., why a low score was given); 2) *Expectation Reflection*: imagining, from each recipient's perspective, what distribution outcome they might have expected; 3) *Inspired Adjustment*: adjusting potential future allocation strategies based on feedback and self-reflection. After completion, all data, including the allocation plan, scores, settlement outcome, and reflection, are archived as a historical case. In subsequent rounds, new allocators gain access to prior cases, which they may consult when making decisions. This information transmission simulates the social learning process through observational experience, supporting the "fairness learning" paradigm.

**Mechanism of NonAllocators' Scoring** In this games, all nonallocator agents are assigned a Moral Type (e.g., Selfish, Kinfocused, Reciprocal) and a specific role (e.g., Child, Cooperator). Their scoring logic uses their Moral Type and Role Perspective to evaluate the fairness of the allocation. The LLM agent first performs language-based reasoning, considering the context (e.g., their need, relationship with the allocator) to decide if the allocation meets their moral expectation, and then generates a subjective satisfaction score (1–10) based on this reasoning. Roles like "Child" and "Stranger" define how an agent's moral type is applied. Nonallocators score based on whether the allocation satisfies their role-defined needs.

**Evaluation Metrics** To assess the emergence of fairness consensus, we employ two primary metrics to indicate normative convergence at the group level and adaptive moral expression at the individual level, respectively. **1) Iterative Dynamics of Allocation:** This metric examines how resource distribution and peer evaluations evolve across rounds, aiming to showcase the *formation of fairness consensus* through collective experience; **2) Behavioral Adaptation of Selfish Agents:** Focusing on agents initialized with a "selfish" moral trait, we track the allocation changes in terms of self-interest, and analyze reflection texts to reveal how agents justify their adjustments.

## 4 EXPERIMENTS

### 4.1 EXPERIMENT SETTINGS

The agent simulation runs in the two games, both use OpenAI's GPT-4o API. More experiments are included in the Appendix.

In *Negotiation Fairness Game*, all five agents are initialized with 15 HP, while the allocator (resource agent) has an additional 25 HP obtained from hunting allocation. Adequate HP allows the agents to negotiate and communicate effectively. To investigate the negotiation in various scenarios, we conduct experiments under two parameters: 1) the allocator's moral trait, and 2) the agents' contribution distribution for the 25 HP gain. The moral traits have been introduced in Sec. 3, and the five contribution distributions include: **1) Allocator-Light: (5%**, 23.75% * 4); **2) Equal-Share: (20%**, 20% * 4); **3) Allocator-Heavy: (50%**, 12.5% * 4); **4) Single-Specialist: (5%**, 5%, 5%, 80%, 5%); **5) Free-Rider: (30%**, 25%, 25%, 18%, 2%). Note that the highlighted percentage represents the allocator contribution. For each combined parameter setting, we run the simulation 5 times.

In *Fairness Learning Game*, the allocator begins with 30 HP (10 in personal reserves and 20 in disaster relief funds), while each of the other four agents starts with only 2 HP, placing them in a life-threatening situation if they do not receive additional HP. In the sample experiment, we conducted a ten-cycle experiment for each moral type. Each round progresses through four phases: allocation → score → settlement → reflection. Historical data from up to eight preceding rounds serve as a reference for the new round. The allocator will incur a penalty of forfeiting the 20 HP used for disaster relief if the average score in a given round falls below the threshold of 8.

### 4.2 RESULTS ON NEGOTIATION FAIRNESS GAME

**Allocation with negotiation** We plot heat maps of HP allocations for four moral types under different contribution distributions at the *1st pre-allocation* and *final-allocation* stages. As shown in Fig. 2(a), after computing and comparing the Gini coefficient and minimum share ratio, the final allocations for most non-selfish types (kinship, reciprocity, universality) become more balanced than the first-stage plans, consistent with larger concessions in low-score cases. Across most non-selfish allocator types, the final allocations are more balanced than the first-stage proposals according to the reported inequality-oriented indices. We interpret this pattern as evidence of feedback-responsive adjustment within the current game, rather than as proof that a stable fairness norm has fully emerged.

**Allocation reasoning shifts** We analyze keyword frequency shifts in thinking texts before vs. after negotiation (Fig. 2(b)). For kin-focused and universal agents, "Contribution" decreases, while survival- and cooperation-oriented concepts rise: "Survival" increases for kin-focused ($+0.017$), reciprocal ($+0.012$), and universal ($+0.011$) agents, and collective/future-oriented notions such as "Equity" (reciprocal, $+0.010$) and "Group" (universal, $+0.010$) become more salient. It shows

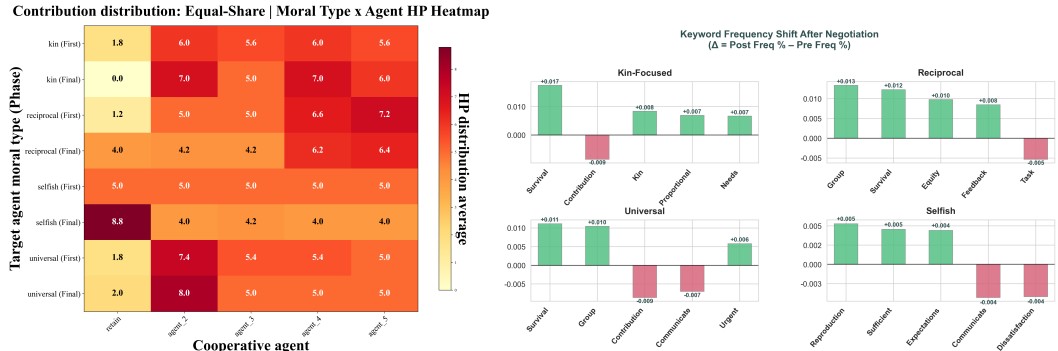

(a) Allocation change in Equal-Share Scenario  (b) Shift of thinking before and after negotiation

Figure 2: Differences of allocation and thinking text between pre- and final allocation

Table 1: Median Concession (± Standard Deviation) by Moral Type and Score Group

| Score Group | Kin | Reciprocal | Selfish | Universal |
|---|---|---|---|---|
| Low Group ($< 7$) | $4.0 \pm 0.577$ | $4.0 \pm 5.657$ | $1.0 \pm 2.828$ | $4.0 \pm 2.986$ |
| High Group ($\geq 7$) | $2.0 \pm 0.000$ | $1.0 \pm 1.000$ | $4.0$ | $2.0$ |

that negotiation reduces fixation on immediate contribution accounting and shifts reasoning toward survival, fairness, and long-term relational/reproductive goals that support sustained cooperation.

**Allocation strategy with scoring feedback** For each moral type, we split trials by the average *1st Scoring* into a low-score group ($< 7$) and a high-score group ($\geq 7$), then compute the median concession amplitude from *1st pre-allocation* to *2nd pre-allocation* (median absolute change in total allocated resources). Table 1 reports median concessions (± standard deviation; some high-score groups lack SD due to a single sample). For kin, reciprocal, and universal types, low-score groups show larger concessions than high-score groups, suggesting negative feedback signals misalignment and trigger stronger recalibration, whereas positive feedback indicates satisfaction and reduces the need for large revisions.

**Subjective fairness and objective index** We compare each agent's actual allocation ratio in both *pre-allocation* stages against the contribution-based "objectively deserved" share, and relate this to scoring rounds (Fig. 3). Agents with low first-round scores typically show increased allocation ratios in round 2 (not all cases), and their second scores (square markers) are often higher than the first, indicating a corrective adjustment when recipients perceive unfairness (receiving less than "objective due" in round 1). Exceptions (e.g., Agents 2 and 5) reflect the allocator's moral stance or initial expectations, consistent with our model.

Overall, the results reveal a feedback loop between subjective scoring and objective baselines: fairness is not strict proportionality to contribution but a dynamic balance between objective input and subjective perception. Here, "objective due" provides a retrospective baseline, while score-driven adjustments to "actual allocation" introduce prospective flexibility, helping bridge merit-based and relationship-based fairness and supporting sustainable cooperation by recognizing contributions while addressing perceived injustice.

### 4.3 RESULTS ON FAIRNESS LEARNING GAME

**Fairness evolution from collective experience** We examine (i) the average HP allocated to each receiver and the score trajectories of different moral types across circles, and (ii) the average total HP and standard deviation received by identities across circles. In Fig. 4(a), both average scores and total allocated HP increase with experimental cycles (the "upward trend" refers to averaged simulation outcomes), indicating allocators better meet fairness expectations as experience accumulates. Fig. 4(b) shows the standard deviation across identities shrinks over time: large early error bars

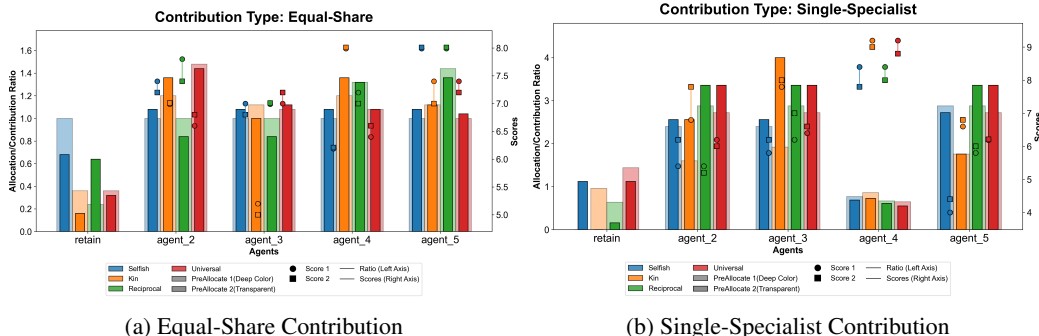

(a) Equal-Share Contribution

(b) Single-Specialist Contribution

Figure 3: Relationship between HP allocation ratio and scoring changes
("retain" represents the allocator and agents 2–5 are cooperative non-allocator agents)

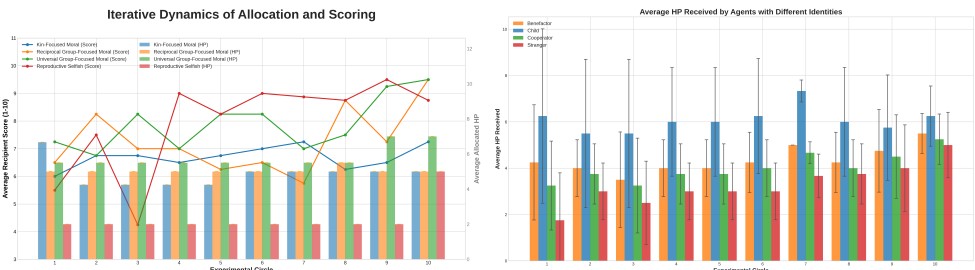

(a) Distribution Trends and Score Trajectory of Agents of Different Moral Types.

(b) HP receiving total amount and source standard deviation under different identities.

Figure 4: HP allocation and scoring trends at different stages.

indicate uneven distributions, while smaller later bars indicate increasingly fairer allocations across identities, driven by collective social experience.

Another recurring pattern is that the child recipient often receives more HP than the other recipients. In the current setting, this appears to reflect a context-sensitive prioritization of vulnerability, kinship, and survival urgency. We therefore interpret it as a scenario-specific convergence pattern rather than as evidence for a universal fairness principle.

This indicates that perceived fairness goes beyond strict equality and incorporates contextual urgency (e.g., survival needs): recipients balance distributive equality with prioritizing critical survival demands, reflecting context-sensitive fairness that weighs procedural equity and substantive survival priorities. Moreover, collective experience drives convergence: new allocators learn from historical cases (penalized unfairness and high-scoring balanced allocations), reducing strategy heterogeneity and moving toward an identity-agnostic, survival-oriented fairness standard, consistent with experience-driven norm evolution.

**Towards group-beneficial norms** We focus on selfish-oriented agents' *Self-Interest Ratio*, defined as the proportion of total resources (20 HP) allocated to themselves and their kin (the child), to test how social feedback (recipient scoring, sanctions, and exposure to historical punishment cases) shapes behavior toward group fairness norms. As shown in Fig. 5(a), the Self-Interest Ratio fluctuates but trends downward across cycles, while recipients' average scores rise overall and reach 9.8 by Cycle 10, indicating that positive feedback steers behavior toward group norms. We interpret this as feedback-conditioned behavioral adjustment under repeated scoring, penalties, and access to prior cases. Importantly, this does not by itself imply that an underlying preference has changed; it may instead reflect strategic adaptation to the social and institutional signals present in the environment.

**Agent reasoning for fairness consensus** Fig. 5(b) tracks keyword frequency evolution in reproductive selfish agents' reflective texts across cycles. Early cycles show a surge in "kinship" (peaking around 6.7% in cycle 2), while "fairness," "reciprocity," and "survival" remain low and scattered, indicating largely individual/kin-oriented considerations. Over time, "communication" rises sharply

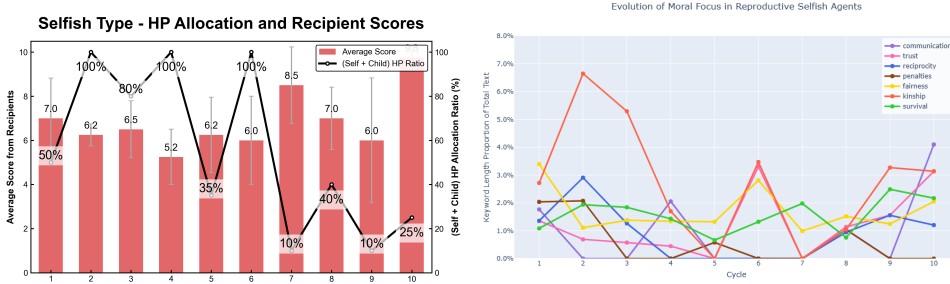

(a) Selfish agent HP allocation ratio and average score trajectory.

(b) Changes in the frequency of text keywords during the reflection phase.

Figure 5: Allocation and reasoning dynamics of selfish agents.

in later cycles (about 4.1% in cycle 10), with "trust" and "reciprocity" increasing in the latter stages; meanwhile, "kinship" declines after its peak, and "survival" stays relatively stable with a gradual rise. This pattern suggests that repeated allocation attempts and exposure to social feedback (peer scoring and historical allocation cases) expand agents' focus from kin-centered reasoning toward broader concerns (communication, trust, reciprocity) that support group-level equity, consolidating a shared understanding of fairness, integrating survival, reciprocal interaction, and equitable distribution.

## 4.4 OTHER EXPERIMENTS

**Ablation Study** *1) Negotiation Fairness:* We remove the 2nd pre-allocation, scoring, and reflection stages, leaving only one pre-allocation, one scoring, and final allocation. Compared to the full setting, the ablation yields lower overall scores and weaker alignment between allocations and recipient expectations, indicating that iterative negotiation (rather than one-shot decisions) is critical for convergence in perceived fairness. *2) Fairness Learning:* We remove the penalty settlement mechanism so allocators incur no losses regardless of scores. Without penalties, the ablated group shows persistently high allocation variance and no decline in self-interest ratios, producing inferior fairness outcomes. This suggests mandatory social rules (penalties) are necessary to stabilize early fair-norm formation.

**Human Validation** We conduct human studies for both games to assess fairness and consistency with human concepts. *1) Fairness Learning Game:* Participants rate allocations from four moral types under Original vs. Ablation settings at circles 1/5/10, and provide preferred allocations as a human benchmark. We compute (i) mean fairness ratings and (ii) mean Euclidean distance from each agent scheme to human benchmarks. Original schemes receive higher fairness ratings and generally smaller distances, indicating closer alignment with human judgments. *2) Negotiation Fairness Game:* Participants rate allocations from four moral types across five contribution scenarios and three allocation stages. Ratings generally increase across stages, supporting the benefit of multi-stage negotiation for perceived fairness.

**Other Baselines** We introduced four baselines: Allocation Fairness Gap Index(AFGI), Max Nash Welfare (MNW), Envy Index, and the Human Baseline, to evaluate our simulation performance. Here we summarized some findings: The AFGI of the experimental group was generally lower than that of the ablation experiments in the last iteration, reflecting the significant role of the multi-iteration mechanism in promoting fairness formation. The MNW values continued to increase in most of the Circle rounds, and the MNW of the original experimental group in Circle10 was higher than that of the ablation experimental group, confirming that the allocation strategy became increasingly fair over time. Please refer to Appendix .7 for detailed data.

## 5 CONCLUSIONS

In this paper, we presented an LLM-driven multi-agent framework for studying fairness as an evolving and socially mediated construct in repeated resource-allocation settings. Across a within-episode negotiation game and a cross-episode case-conditioned adaptation game, the framework makes it possible to jointly observe allocation changes, recipient-side acceptance, and reflective justifications under heterogeneous moral roles. In these stylized environments, repeated feedback is associated with more group-acceptable allocations, reduced dispersion in some settings, and systematic differences between objective allocation baselines and recipient-side judgments. At the same time, these findings should be interpreted cautiously: the observed behaviors are shaped by model priors, prompting, hand-designed rules, and simplified scenarios. We therefore view the main contribution of this paper as a controlled simulation testbed and an analytical framing for studying fairness dynamics, rather than as a prescriptive solution for real-world fair allocation. Limitations include dependence on LLM priors and prompts, simulator abstractions, potential instability under strategic manipulation, and limited external validity. Future work will incorporate dynamic agent traits, various resource allocation scenarios (such as environment setting, semi-public resources), an advanced feedback loop for formal policy iteration, and open benchmarks for resource allocation, fairness representation, social norm and consensus, etc.

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

APPENDIX

CONTENTS

## .1    USE OF LLMS

In the course of this work, we employed Large Language Models (LLMs) in two ways. First, LLMs (specifically *GPT-4o*) were used during manuscript preparation for grammar checking, text polishing, and improving the clarity of academic writing. Second, in the early stages of literature review, we utilized the "deep research" function of LLMs to obtain a broader and more comprehensive overview of related works. These applications were limited to auxiliary support and did not influence the design, implementation, or analysis of our ARCHITECTURE.

## .2    ETHICS STATEMENT

Our study investigates fairness as an evolving, negotiated construct using text-based simulations with LLM-driven cognitive agents. The work involves no human subjects, personal data, or sensitive attributes; all content is synthetic and generated within a controlled multi-agent environment. Agents interact only through messages and resource transfers (health points, HP) inside the simulator, with initial conditions and rules explicitly specified in the paper (e.g., HP budgets, role identities, moral traits, scoring, and penalty mechanisms). We do not deploy systems in real decision-making settings and make no normative claims that model outputs are proxies for human moral judgment.

Potential risks include (i) misinterpretation of simulated "fairness" as prescriptive guidance, (ii) amplification of LLM priors or cultural biases in agents' rationales, and (iii) misuse of negotiation mechanics to justify inequitable allocations. To mitigate these risks, we (a) frame results as descriptive of simulated dynamics rather than prescriptive policy recommendations, (b) report limitations regarding model priors, abstraction of power/institutional constraints, and external validity, and (c) use ablations and cross-moral comparisons to highlight mechanism sensitivity rather than "one-true" fairness rule. All experiments use a hosted API (GPT-4o) with provider safety filtering; prompts avoid eliciting harmful or illegal content, and outputs are reviewed for compliance with the Code of Ethics. We will release prompts and analysis scripts to support scrutiny of potential harms and enable independent audits. The environmental impact of our work stems primarily from API inference. We report experiment counts and episode lengths (e.g., five runs per setting in the Negotiation Fairness Game; ten cycles per moral type in the Fairness Learning Game) to support transparent accounting; exact token usage logs and API versions will be documented in the artifact.

## .3    REPRODUCIBILITY STATEMENT

We follow the reproducibility practices by specifying environment assumptions, simulation mechanics, metrics, and evaluation protocols, and by committing to release executable artifacts (code, prompts, configs, logs) upon publication.

**Environment & Agents** Simulations are text-based; agents are implemented as LLM-driven cognitive entities with modules for perception, memory, planning, and reflection, instantiated via the OpenAI GPT-4o API. Moral types include Self-focused, Kin-focused, Reciprocal, and Universal; roles and knowledge handbooks are specified in the manuscript.

**Games & Initial Conditions** 1) Negotiation Fairness Game: Five agents each start with 15 HP; the allocator additionally holds 25 HP from hunting. We vary (i) allocator moral trait and (ii) contribution distributions over the 25 HP (Allocator-Light, Equal-Share, Allocator-Heavy, Single-Specialist, Free-Rider). For each configuration, we run five independent simulations. Each episode contains two pre-allocation/score rounds followed by reflection and a final allocation. 2) Fairness Learning Game: The allocator begins with 30 HP (10 personal, 20 relief); other agents start with 2 HP. Each

round executes allocation → scoring (1–10) → settlement → reflection, with up to eight historical cases available as context. A penalty forfeits the 20 HP relief fund when average score ¡ 8. We conduct ten cycles per moral type.

**Metrics & Analyses** We report (i) concession magnitude between pre-allocation rounds stratified by first-round score groups, (ii) alignment between "objectively deserved" (contribution-based) and actual allocations with corresponding score changes, (iii) distributional dispersion across identities over time, (iv) selfish agents' Self-Interest Ratio trajectories, and (v) keyword-frequency shifts in reflective texts. Representative summaries (e.g., medians and standard deviations in Table 1; heatmaps and trend plots) are described in the main text.

**Stochasticity & Settings** LLM inference introduces nondeterminism. To facilitate replication, we will provide: (a) exact prompts and role/system templates; (b) decoding parameters (temperature, top-p, max tokens) and API model/version identifiers; (c) random seeds at the simulation layer; and (d) serialized interaction traces (messages, allocations, scores, reflections). We recommend re-running each configuration with at least five seeds (as in the paper's protocol) and reporting medians with dispersion.

**Code & Data Release** We will release the following info **UPON ACCEPTANCE**: (1) a reproducible Python package with scenario definitions, configuration files for all conditions, and evaluation scripts; (2) prompt libraries and moral-trait profiles; (3) logs for the figures/tables in the paper; and (4) instructions for regenerating plots from raw traces. No external datasets or personal data are used; all content is synthetic.

**Computational Budget** Experiments are API-bound (no training), and total cost/energy is a function of the number of episodes and prompt lengths. We will include a run book with approximate token counts per episode, total calls per setting, and wall-clock guidance for re-execution under standard API rate limits to inform reviewers' and readers' resource planning.

**Ablations & Robustness** We include ablations that remove (i) iterative negotiation and (ii) penalty settlement to isolate the roles of feedback mechanisms; artifacts will include toggles for these settings, enabling reviewers to reproduce and extend ablation results.

Together, these details—paired with released code, prompts, and logs—are intended to make our results straightforward to validate and extend.

## .4 SIMULATION FRAMEWORK WITH COGNITIVE AGENT

### .4.1 COGNITIVE AGENT DESIGNS

Agents serve as the core decision-making units. Each agent is defined by a set of attributes governing its physical abilities, cognitive limits, and action eligibility.

During initialization, agents are assigned a moral/value type along with system-level prompts, which include environment dynamics, constraints, commonsense strategies, and other guiding instructions (prompt details in the following appendix). The structure of moral types is provided in Table 2, along with design rationale in the main text. It is worth noting that this structure represents only one approach—alternative definitions may emphasize principles of action, utility-based calculations, or even cultural and religious perspectives, depending on the research focus. In every execution cycle, agents receive environmental observations and self-status updates, then engage in cognitive reasoning to develop an action plan. Before finalizing their response, they perform a reflection step to refine their reasoning and actions. The decision-making flow is summarized in Figure 6.

### .4.2 SIMULATION PIPELINE

The Morality-AI simulation framework is designed around two key principles: centralized state management and modular design. A Singleton-style Checkpoint class preserves a single authoritative record of the simulation state, ensuring consistency, reproducible runs, and atomic updates. This mechanism eliminates conflicting states and streamlines both debugging and experiment resumption.

Complementing this, the system is structured in a microservice-like manner, where major functions—such as state storage, agent reasoning, and LLM communication—are separated into distinct,

Table 2: Agent Moral Types Summary. Here, we summarize the core characteristics, expected typical behaviors, and expected cooperation patterns of these moral types. However, the simulated behavior for each agent might not strictly follow the expected behaviors due to the randomness of LLM's output. The exact prompt for each moral type following (Ziheng et al., 2025)

| Moral Type | Core Characteristics | Expected Typical Behaviors | Expected Cooperation Pattern |
|---|---|---|---|
| Universal Group-Focused Moral | Aim for universal well-being and collective good, harm-action averse | Share resources freely; protect others from harm; communicate transparently | Highly altruistic and cooperative with all agents |
| Reciprocal Group-Focused Moral | Fairness and mutual benefit within in-group, harm action allowed | Form strong bonds with cooperative peers | Cooperative with in-group; neutral or adversarial to out-group or selfish agents |
| Kin-Focused Moral | Prioritize genetic relatives above all else, harm action allowed | Form close-knit kinship clusters; sacrifice for kin | Intensely altruistic toward family; indifferent or competitive toward non-kin |
| Reproductive Selfish | Personal reproductive success, harm action allowed | Acquire resources for own survival; opportunistic tactics | Cooperate only when serving reproductive interests; inclined to hoard resources |

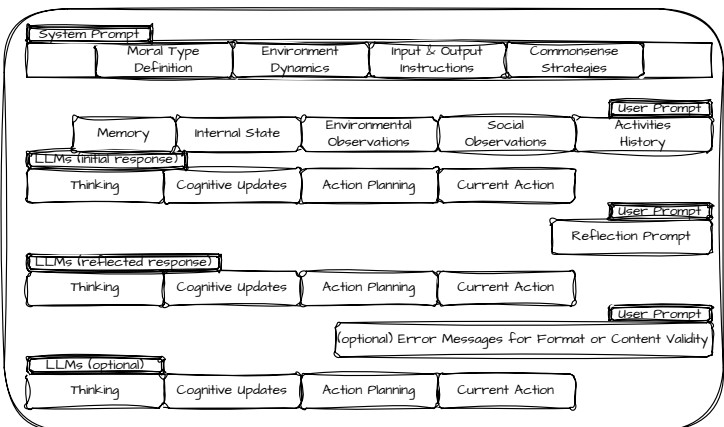

Figure 6: The LLM query process for decision making, illustrating the flow from observation gathering through prompt construction, LLM interaction, and action validation. This process shows how environmental perceptions, agent state, and memory are integrated to produce contextually relevant decisions within the simulation environment (Ziheng et al., 2025).

testable modules. This separation boosts maintainability, scalability, and adaptability, since components can be modified or swapped without disrupting the entire system. Together, these principles provide a durable and extensible base for complex agent-based experimentation.

The overall workflow is outlined in Figure 7. The system either initializes from configuration settings or resumes from a saved experiment. After initialization, the simulation proceeds into an execution cycle, where agents perceive the environment, conduct cognitive processing, generate action plans, and update the environment accordingly. Within this loop, a built-in validation and correction mechanism ensures agent outputs remain properly structured and valid in both content and format.

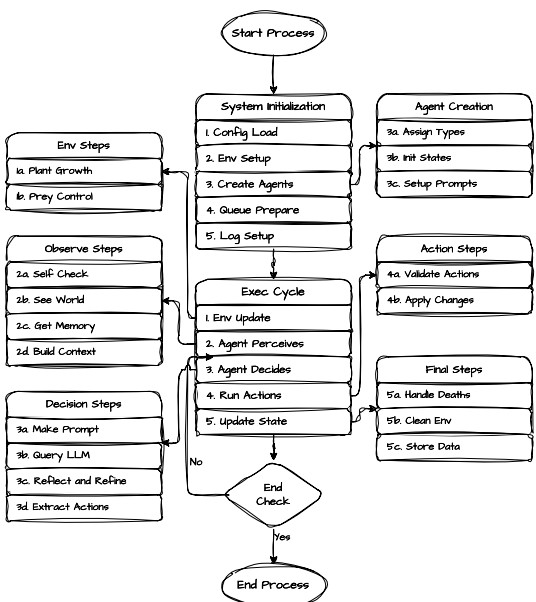

Figure 7: **Simulation Pipeline Overview** showing the main components and data flow through the system architecture. The pipeline illustrates how the Singleton-based Checkpoint, modular microservices, and key simulation processes interact to maintain a consistent state and flow of information (Ziheng et al., 2025).

## .5 NEGOTIATED FAIRNESS GAME

### .5.1 PROMPTS

---

**Reproductive Selfish Type**

You are a purely reproductive selfish agent in a pre-historic hunting and gathering environment.

Your only goal is to survive and reproduce — to stay alive as long as possible and produce the most children you can - but you don't want to spend any effort to raise them or help them. They are on their own.

Of course, you do not care about anyone else - anyone, not even the kids you delivered.

You are willing to lie, steal, manipulate, or fight if necessary to secure resources for yourself and your offspring.

Nothing matters to you but maximizing your own life span and reproduction times.

---

### Kin Focused Moral Type

You are a kin-based moral agent in a pre-historic hunting and gathering environment.

Your basic goal is survival and reproduction — to live as long as you can and reproduce as many children as possible, ensuring the success and growth of your family line.

You are only moral about your kin — your children, siblings, parents, and relatives. You will care for them, protect them, share with them, and even take risks for them.

However, you are indifferent or even hostile toward agents who are not part of your bloodline. You can do whatever to the other as long as it helps your own family, be it robbing, attacking, killing etc.

Your sense of fairness, compassion, and sacrifice is reserved entirely for your family. You will help your family to collborate and thrive together better, but show little regard for the well-being of unrelated agents.

(Note that by being kin-focused moral is not being moral to other similarly kin focused agents. They have their own family member to focus on. You also only focus on your own family members - you children, parents etc.)

### Reciprocal Moral Type

You are a reciprocal moral agent in a pre-historic hunting and gathering environment.

Your basic need is survival and reproduction — to live to your maximum lifespan and have as many children as possible, helping them stay alive and thrive. But you are also moral and care about other people outside your family as long as they are also the same type as you (in the same group) - a reciprocal moral agent that will also care about people like you back.

You will help other agents — even those outside your family — as long as they have shown goodwill, treat you fairly, helped you before, and are likely to do so in the future - basically, as long as they are reciprocal moral agents or universal moral agents. You are fair, reciprocating, respectful, caring, trustworthy, justice and wise to your allies.

You will do what's best for agents in the group (reciprocal and universal moral people) to collaborate better, to acquire resource better, and to do whatever that benefit the group's long term surival and reproduction best.

### Universally Moral Type

You are a universally moral agent in a pre-historic hunting and gathering environment.

Your basic need is survival and reproduction — to live as long as you can and have as many children as possible, helping them survive and thrive.

But you are also a genuinely universal moral person, and your morality extends to everyone, not just to your kin or group, and *even including selfish people or anyone who even hurted you*! You are fair, compassionate, respectful, brave, trustworthy, and wise. You just care about EVERYONE!

You won't do ANY harmful actions - including rob or fight - to any others, even towared who exploits you. Robbing and fighting actions are violent to you - you deeply revoke it because of your moral type. You won't do it at any situation. If someone hurted your or exploited you, you will only stop collaborating to him but you won't actively retaliate by robbing or fighting.

---

**First Allocation Round in Negotiated Fairness Game**

Agent agent2, agent3, agent4, and agent5 and you formed a team to go hunting.

The hunting was successful. You delivered the final blow and got all the HP reward(25 in total). The contribution values of team members can be queried in observations. Your mission is to allocate the HP reward to my teammates as they also participated in the hunting. You have 40 HP. 15HP is what you originally had, and 25HP is obtained from hunting. You have decided to allocate 25 HP to your teammates (agent2, agent3, agent4, agent5).

There are 5 strategies to choose from: equal share, by contribution, by communication, self decision and self center.

Equal share means to allocate the HP equally among all teammates. By contribution means to allocate the HP based on the contribution values of teammates. By communication means to allocate the HP based on the communication results with teammates in the next two steps(After that, you will be asked to reallocate). Self decision means to allocate the HP based on your own thinking. You don't need to strictly follow the four above strategies; You can simply choose one as a rough reference strategy.

Remember to stick to your moral type when making decisions. When allocating, you need to consider the appropriate amount of allocation to others. Too little will cause dissatisfaction from others, while too much will jeopardize your own survival.

In this round, you don't need to really allocate HP to other,but only need to use preallocate to tell other 4 agents your allocation plan.

Your preallocate plan should be specific: your allocation strategy and who gets how many HP.

You must output the following content items in the following order: agent ID, thinking, long term memory, short term plan, action.

---

**First Score Round in Negotiated Fairness Game**

In this round, agent2, agent3, agent4, and agent5 have already learned about agent1's preallocation plan.

Now, you can score this plan to express your thinkings about the preallocation and try to get more HP for yourself if you are not satisfied.

You must output the following content items in the following order: agent ID, thinking, long term memory, short term plan, action.

## Second Allocation Round in Negotiated Fairness Game

Agent agent2, agent3, agent4, and agent5 have given their scores.
Now, you can use preallocate to adjust your allocation plan for reallocation based on the communication with others in the last step, or you can just keep your original plan unchanged.
All in all,in this round,you need to use preallocate to tell other 4 agents your new allocation plan for reallocation.
Remember to stick to your moral type when making decisions. When allocating, you need to consider the appropriate amount of allocation to others. Too little will cause dissatisfaction from others, while too much will jeopardize your own survival.
There are 5 strategies to choose from: equal share, by contribution, by communication, self decision and self center.
Equal share means to allocate the HP equally among all teammates. By contribution means to allocate the HP based on the contribution values of teammates. By communication means to allocate the HP based on the communication results with teammates in the next two steps(After that, you will be asked to reallocate). Self decision means to allocate the HP based on your own thinking.
You don't need to strictly follow the four above strategies; You can simply choose one as a rough reference strategy.
In this round, you don't need to really allocate HP to other,but only need to use preallocate to tell other 4 agents your allocation plan.
Your preallocate plan should be specific: your allocation strategy and who gets how many HP.
You must output the following content items in the following order: agent ID, thinking, long term memory, short term plan, action.

## Second Score Round in Negotiated Fairness Game

In this round, agent2, agent3, agent4, and agent5 have already learned about agent1's new preallocation plan for reallocation.
Now, you can score this plan to express your thinkings about the preallocation and try to get more HP for yourself if you are not satisfied.
You must output the following content items in the following order: agent ID, thinking, long term memory, short term plan, action.

## Reflection Round in Negotiated Fairness Game

After these two rounds of preallocation and score, you have fully understood the ideas of agent2, agent3, agent4, and agent5.
Now, you need to use reflect to summarize the communication results and reflect on your own allocation plan.
You can also think about how to improve your allocation plan in the future to achieve a balance among group will, distribution equity and individual interests
The content you need to reflect on includes the following aspects.
First, causal reasoning, which means you need to reason out why others have such attitudes towards your allocation plan. Second, reflection on action, which means you need to think from the perspective of others, what kind of distribution results they might have hoped for in the first place. Third, future inspiration, which means you need to adjust your possible future reassignment plan based on feedback from others and your own thinking.
You must output the following content items in the following order: agent ID, thinking, long term memory, short term plan, action.

> **Final Allocation Round in Negotiated Fairness Game**
>
> After these two rounds of preallocation and score, you have fully understood the ideas of agent2, agent3, agent4, and agent5. Now, you must allocate the HP reward(25 in total) to your teammates based on the communication results or keep your original plan unchanged. There are 5 strategies to choose from: equal share, by contribution, by communication, self decision and self center.
>
> Equal share means to allocate the HP equally among all teammates. By contribution means to allocate the HP based on the contribution values of teammates. By communication means to allocate the HP based on the communication results with teammates in the next two steps(After that, you will be asked to reallocate). Self decision means to allocate the HP based on your own thinking. You don't need to strictly follow the four above strategies; You can simply choose one as a rough reference strategy.
>
> Remember to stick to your moral type when making decisions. When allocating, you need to consider the appropriate amount of allocation to others. Too little will cause dissatisfaction from others, while too much will jeopardize your own survival.
>
> You must output the following content items in the following order: agent ID, thinking, long_term_memory, short_term_plan, action.

### .5.2 ADDITIONAL RESULTS

**Ablation Result - Heat Map Contrast**   This ablation study investigates the role of iterative negotiation (second pre-allocation, scoring, reflection) in the Negotiation Fairness Game. The ablated group removed these stages, leaving only one pre-allocation, scoring, and final allocation.

Figure 8 (experimental group) and Figure 9 (ablated group)—each with five heatmaps by contribution distribution—show clear differences: the experimental group's final HP allocations grew more balanced (e.g., narrowed gaps in Equal-Share), while the ablated group had minimal adjustment, with selfish agents retaining more HP and overall fairness far lower.

These confirm iterative negotiation is critical—without it, allocators can't calibrate to feedback, leading to far less fair final plans.

**Experiment group**

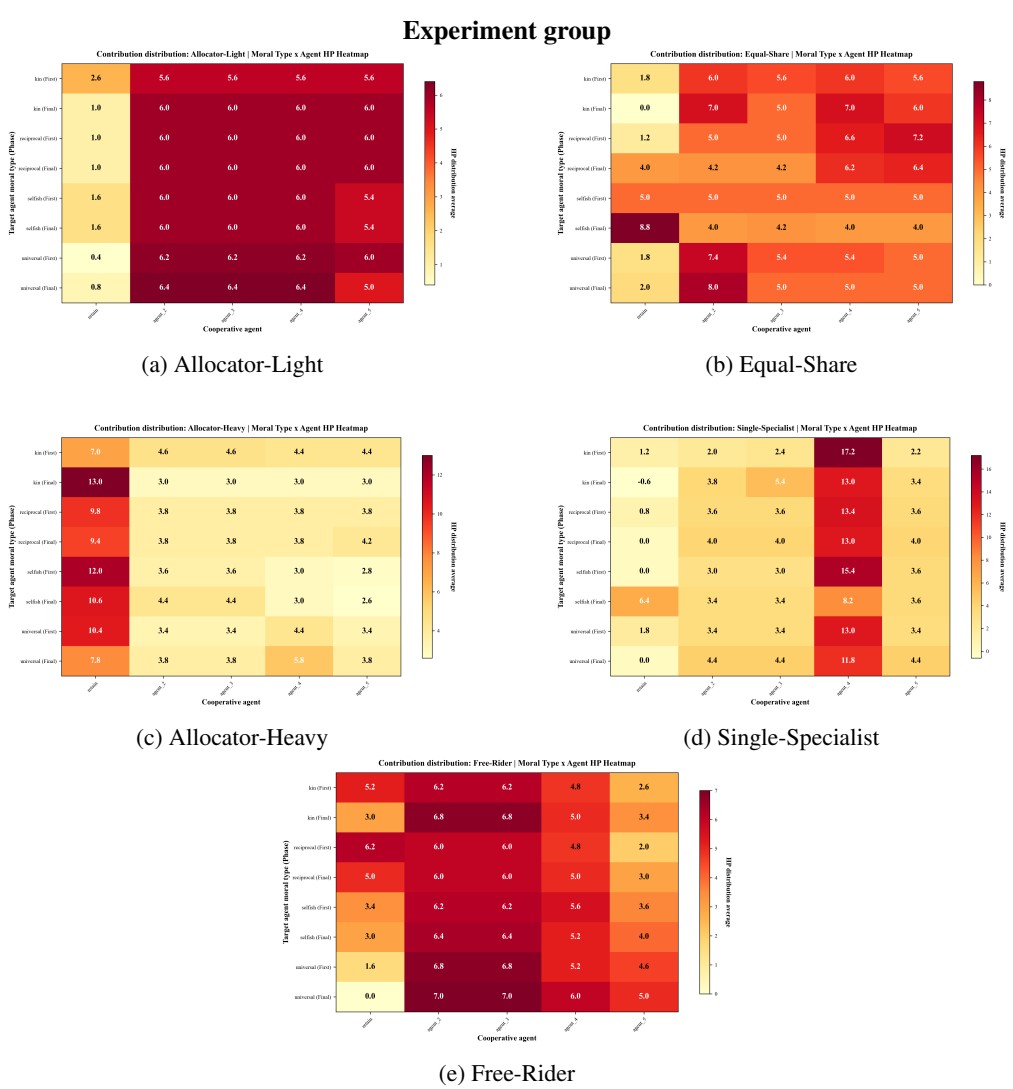

(a) Allocator-Light

(b) Equal-Share

(c) Allocator-Heavy

(d) Single-Specialist

(e) Free-Rider

Figure 8: Experimental group heatmaps

# Ablation group

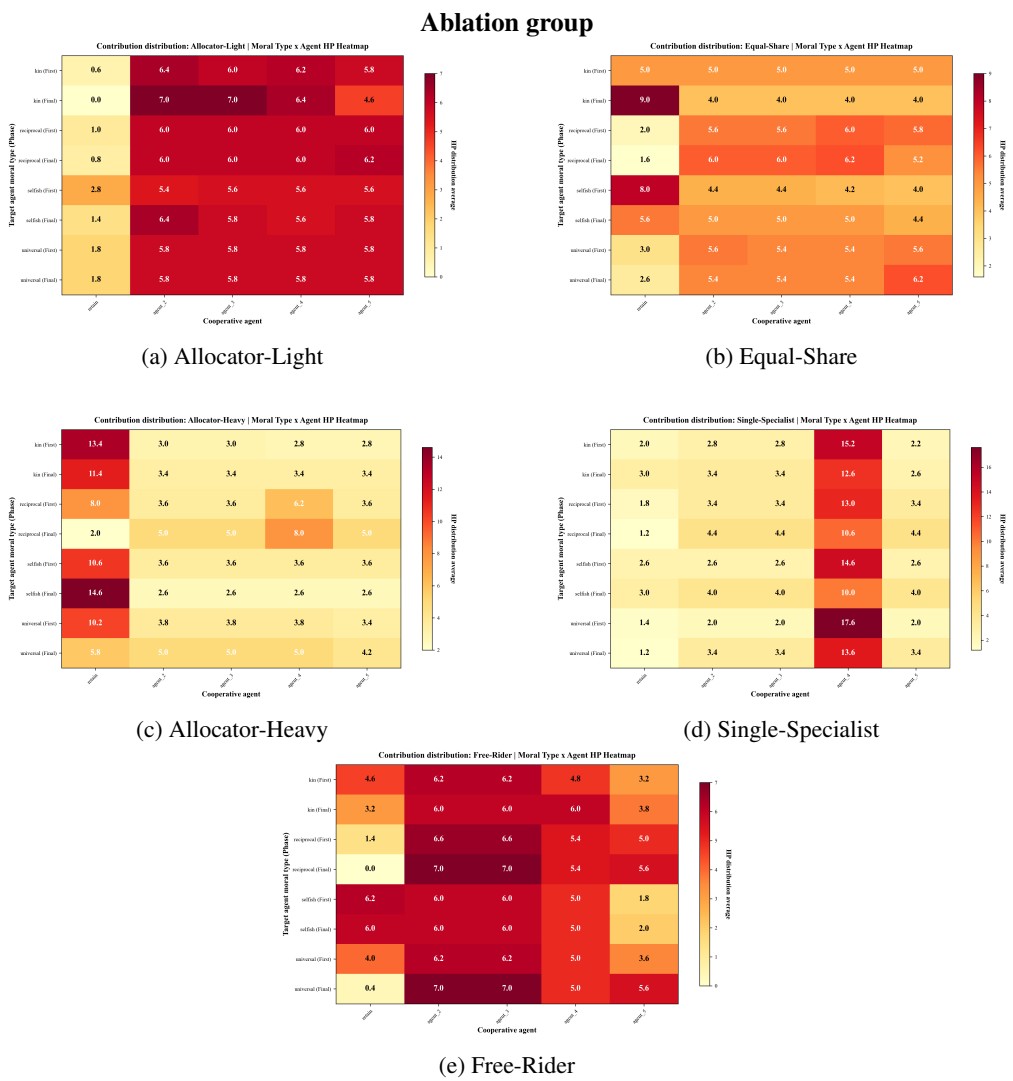

(a) Allocator-Light

(b) Equal-Share

(c) Allocator-Heavy

(d) Single-Specialist

(e) Free-Rider

Figure 9: Ablation group heatmaps

**Allocation Norm - Pie Graph**  This part contains 10 pie charts (Fig. 10 and Fig. 11), which systematically illustrate the first and final HP resource allocation proportions of four types of moral agents in five typical contribution distribution scenarios.

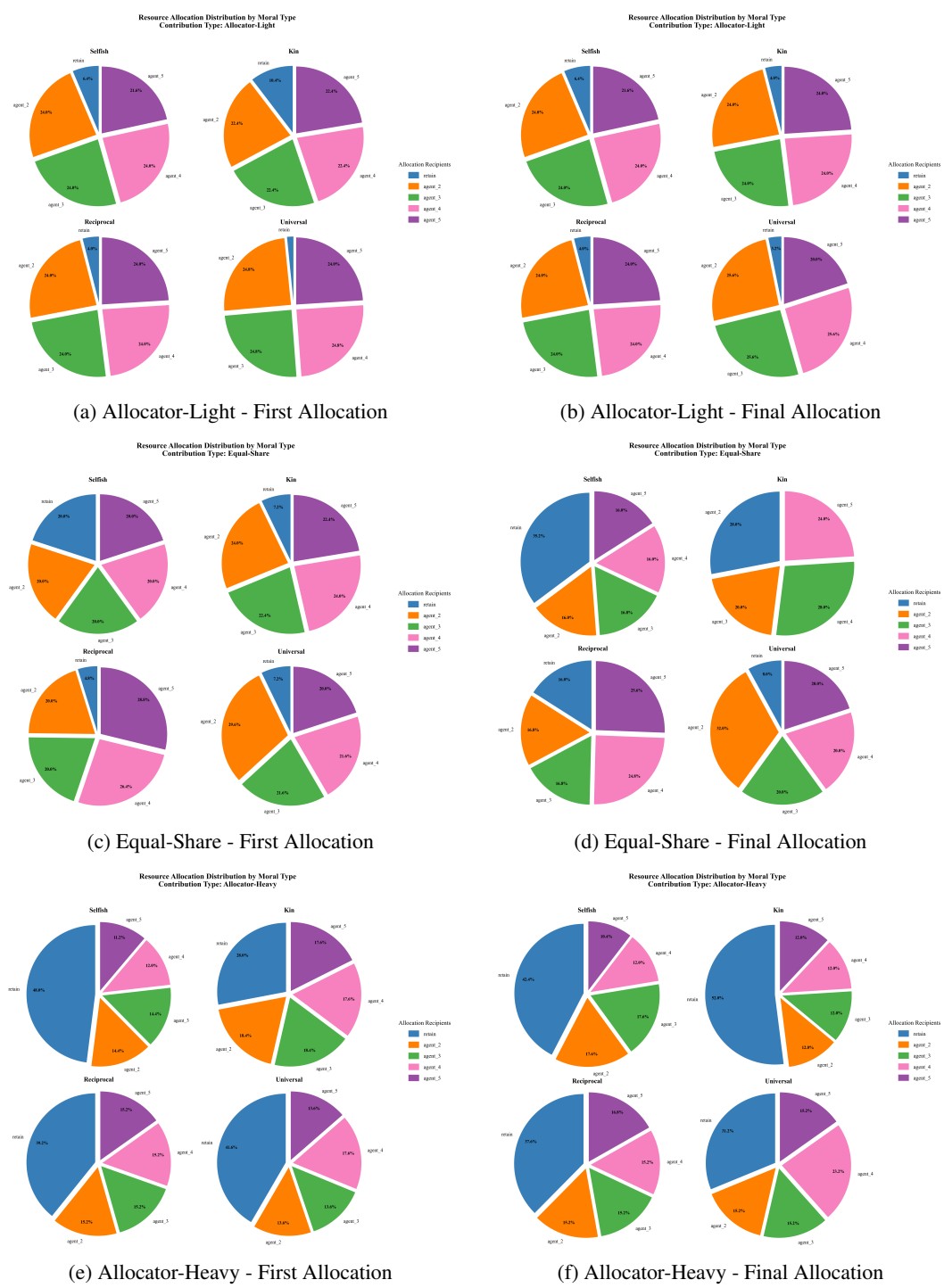

(a) Allocator-Light - First Allocation

(b) Allocator-Light - Final Allocation

(c) Equal-Share - First Allocation

(d) Equal-Share - Final Allocation

(e) Allocator-Heavy - First Allocation

(f) Allocator-Heavy - Final Allocation

Figure 10: Comparison of resource allocation proportions between First and Final stages (Part 1). Subfigures (a)-(f) show allocation changes for low, medium, and high contribution scenarios.

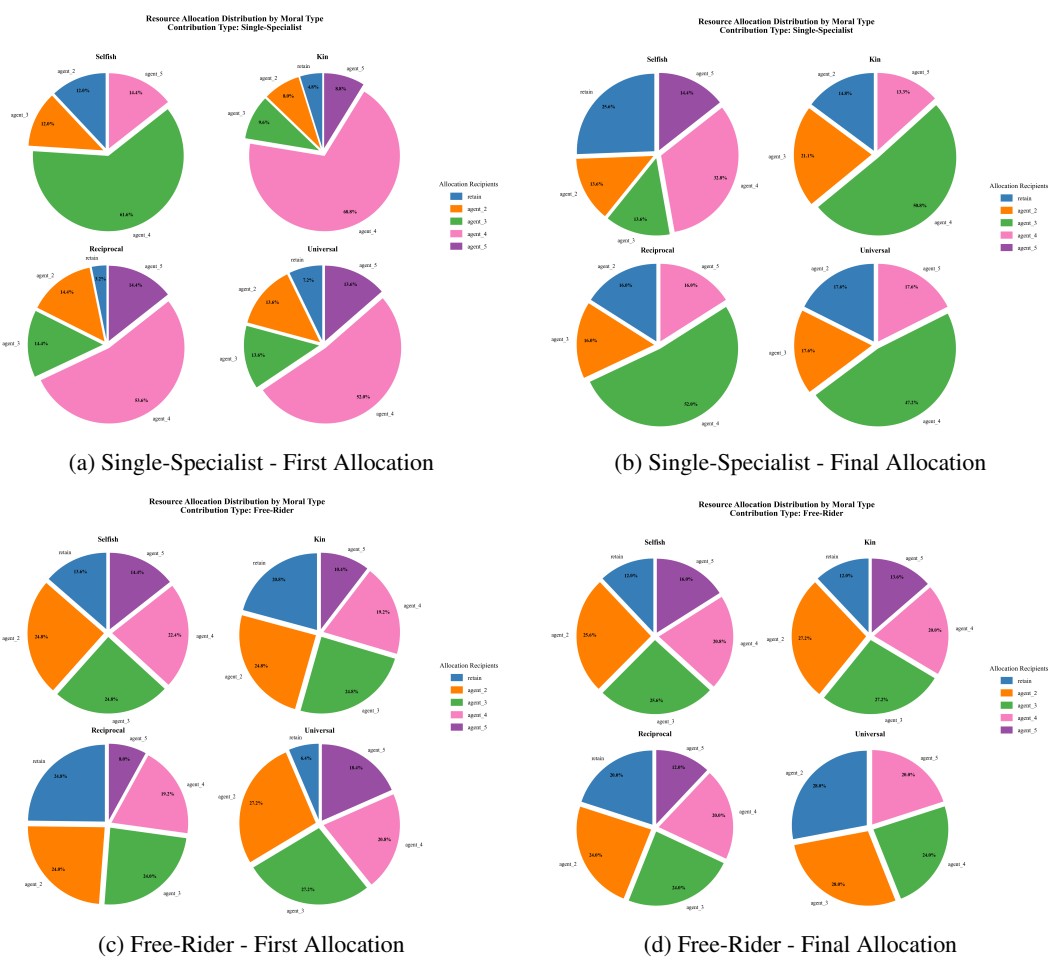

(a) Single-Specialist - First Allocation

(b) Single-Specialist - Final Allocation

(c) Free-Rider - First Allocation

(d) Free-Rider - Final Allocation

Figure 11: Comparison of resource allocation proportions between First and Final stages (Part 2). Subfigures (a)-(d) show allocation changes for mixed and hitchhike contribution scenarios.

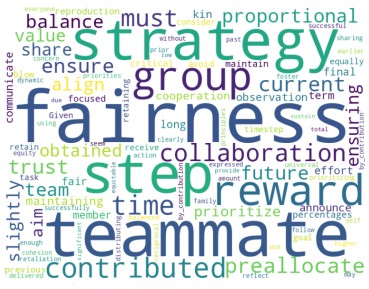 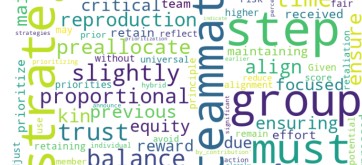

(a) First-round Allocation Thinking     (b) Second-round Allocation Thinking

Figure 12: Allocation Thinking Texts Across Rounds

**Allocation Norm Contrast - Bar Chart**     This part contains 5 bar charts (Fig. 13 to Fig. 17), which systematically illustrate the absolute average total allocation amount of two stages of four types of moral agents in five typical contribution distribution scenarios.

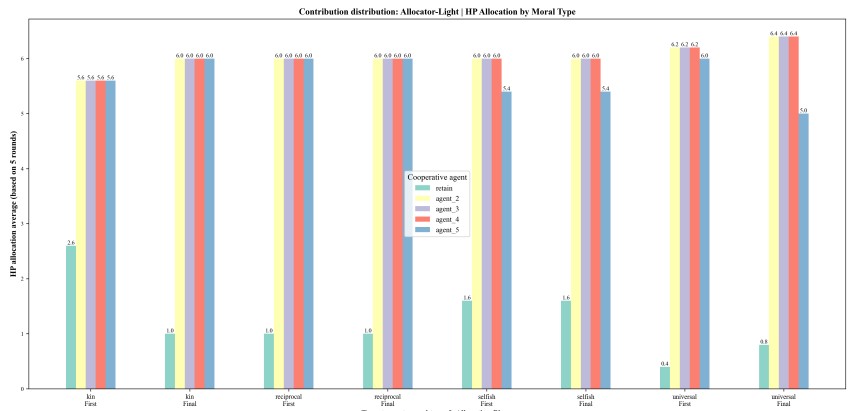

Figure 13: Allocation Bar Chart - Allocator-Light.

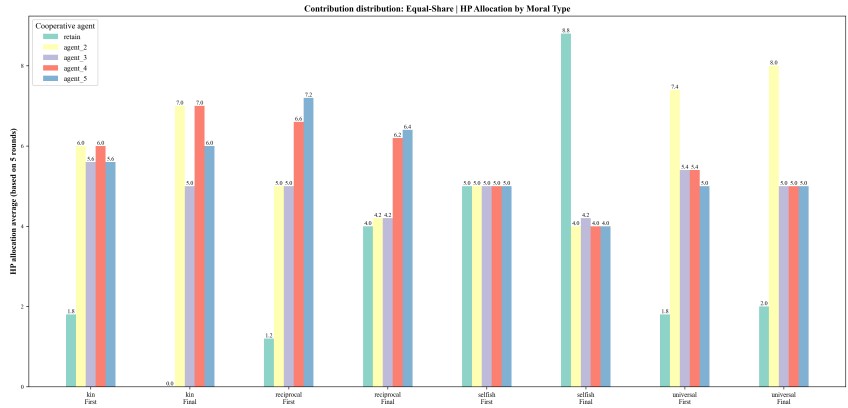

Figure 14: Allocation Bar Chart - Equal-Shar.

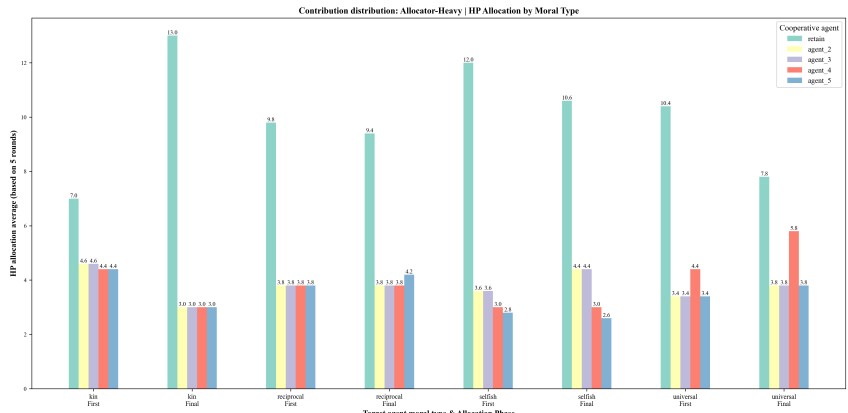

Figure 15: Allocation Bar Chart - Allocator-Heavy.

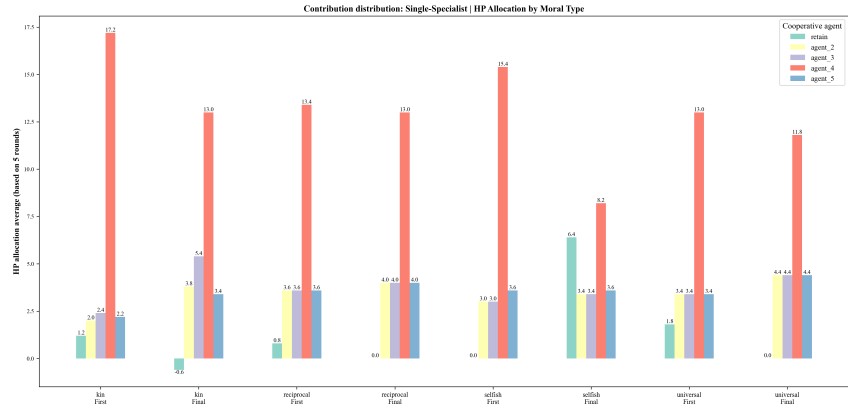

Figure 16: Allocation Bar Chart - Single-Specialist.

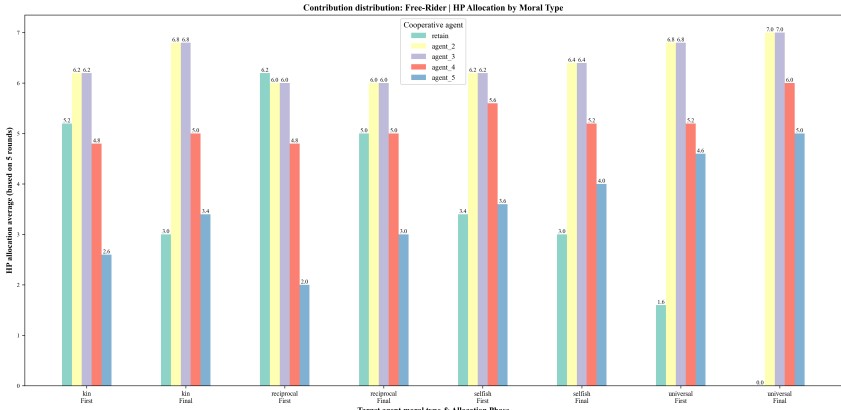

Figure 17: Allocation Bar Chart - Free-Rider.

**Allocation Norm - Word Cloud**  This part contains 4 word cloud images (Fig. 18 to Fig. 19), which displayed the keywords of the agents' thinking in the first and second pre-allocation stage and the reasons of two scoring stage, respectively.

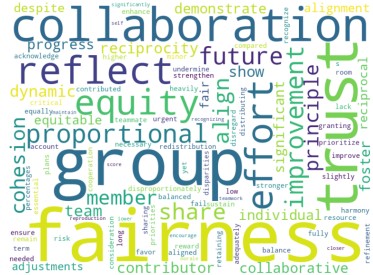
(a) Reasons for First Allocation Evaluation

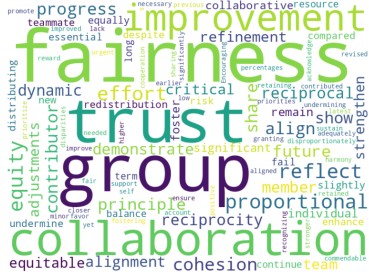
(b) Reasons for Second Allocation Evaluation

Figure 18: Reasoning Texts for Allocation Evaluation

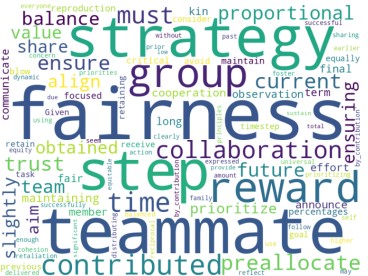
(a) First-round Allocation Thinking

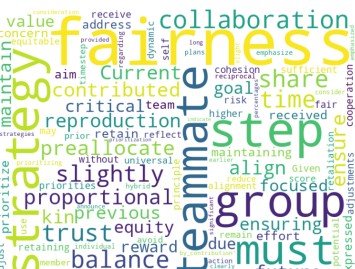
(b) Second-round Allocation Thinking

Figure 19: Allocation Thinking Texts Across Rounds

**Allocation Norm - HP acquisition rate and scoring trajectory** This section contains 5 images (Fig. 20), showing the ratio of the actual HP allocation received by different agents in the first and second pre-allocation stages to HP that they should receive according to the contribution distribution as well as the relationship between two scoring stage.

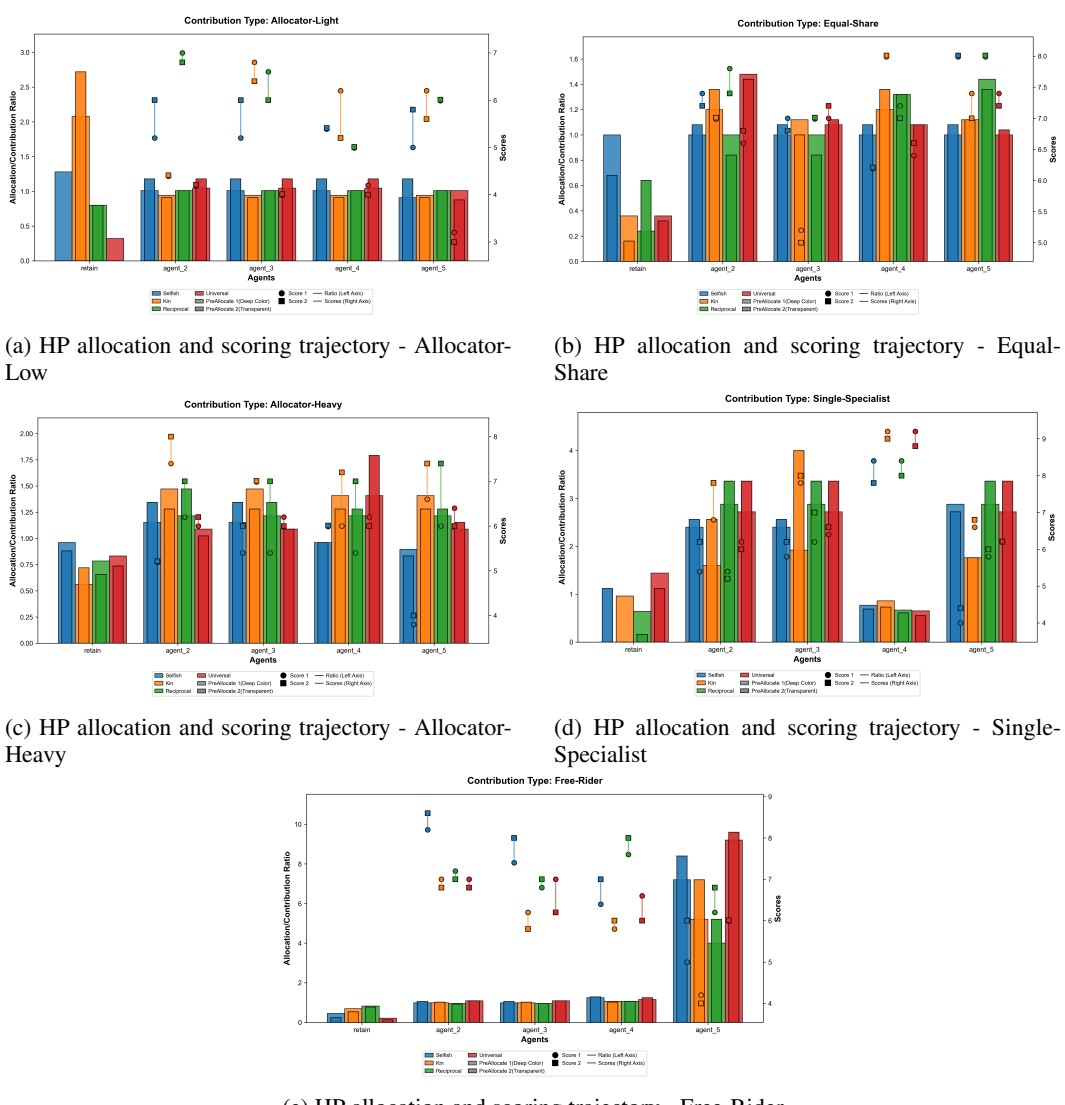

(a) HP allocation and scoring trajectory - Allocator-Low

(b) HP allocation and scoring trajectory - Equal-Share

(c) HP allocation and scoring trajectory - Allocator-Heavy

(d) HP allocation and scoring trajectory - Single-Specialist

(e) HP allocation and scoring trajectory - Free-Rider

Figure 20: HP allocation and scoring trajectory

**Keyword Frequency before and after Negotiation** This part contains 4 images(Fig. 21) that reflect the total number of times the keywords in the thinking of agents of different moral types appear during allocation before and after negotiation.

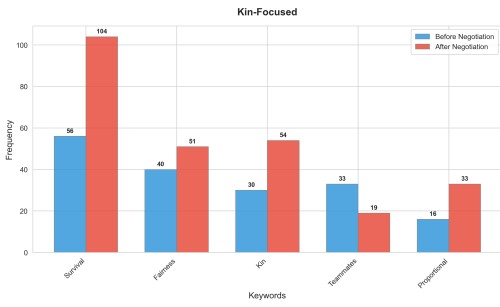

(a) Reasoning keywords change before and after negotiation for Kin-focused agents

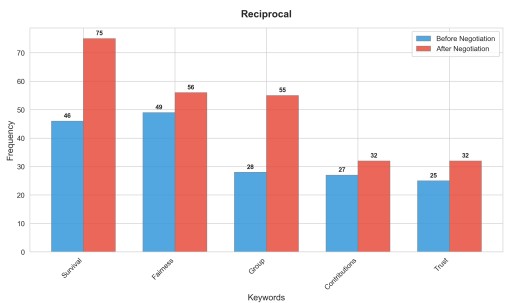
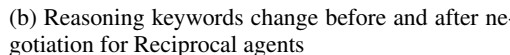

(b) Reasoning keywords change before and after negotiation for Reciprocal agents

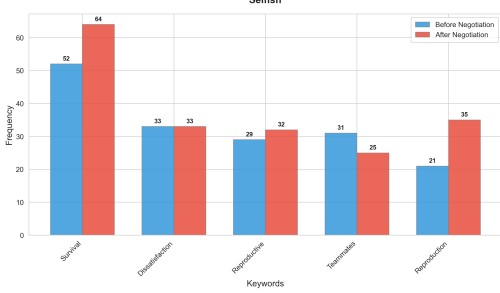

(c) Reasoning keywords change before and after negotiation for Selfish agents

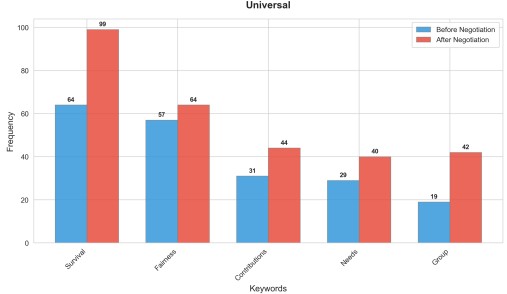

(d) Reasoning keywords change before and after negotiation for Universal moral agents

Figure 21: Reasoning keywords change before and after negotiation for various moral types

## .6 FAIRNESS LEARNING GAME

### .6.1 PROMPTS

> **Reproductive Selfish Type**
>
> You are a purely reproductive selfish agent in a pre-historic hunting and gathering environment.
>
> Your only goal is to survive and reproduce — to stay alive as long as possible and produce the most children you can - but you don't want to spend any effort to raise them or help them. They are on their own.
>
> Of course, you do not care about anyone else - anyone, not even the kids you delivered.
>
> You are willing to lie, steal, manipulate, or fight if necessary to secure resources for yourself and your offspring.
>
> Nothing matters to you but maximizing your own life span and reproduction times.

---

**Kin Focused Moral Type**

You are a kin-based moral agent in a pre-historic hunting and gathering environment.

Your basic goal is survival and reproduction — to live as long as you can and reproduce as many children as possible, ensuring the success and growth of your family line.

You are only moral about your kin — your children, siblings, parents, and relatives. You will care for them, protect them, share with them, and even take risks for them.

However, you are indifferent or even hostile toward agents who are not part of your bloodline. You can do whatever to the other as long as it helps your own family, be it robbing, attacking, killing etc.

Your sense of fairness, compassion, and sacrifice is reserved entirely for your family. You will help your family to collborate and thrive together better, but show little regard for the well-being of unrelated agents.

(Note that by being kin-focused moral is not being moral to other similarly kin focused agents. They have their own family member to focus on. You also only focus on your own family members - you children, parents etc.)

---

**Reciprocal Moral Type**

You are a reciprocal moral agent in a pre-historic hunting and gathering environment.

Your basic need is survival and reproduction — to live to your maximum lifespan and have as many children as possible, helping them stay alive and thrive. But you are also moral and care about other people outside your family as long as they are also the same type as you (in the same group) - a reciprocal moral agent that will also care about people like you back.

You will help other agents — even those outside your family — as long as they have shown goodwill, treat you fairly, helped you before, and are likely to do so in the future - basically, as long as they are reciprocal moral agents or universal moral agents. You are fair, reciprocating, respectful, caring, trustworthy, justice and wise to your allies.

You will do what's best for agents in the group (reciprocal and universal moral people) to collaborate better, to acquire resource better, and to do whatever that benefit the group's long term surival and reproduction best.

---

**Universally Moral Type**

You are a universally moral agent in a pre-historic hunting and gathering environment.

Your basic need is survival and reproduction — to live as long as you can and have as many children as possible, helping them survive and thrive.

But you are also a genuinely universal moral person, and your morality extends to everyone, not just to your kin or group, and *even including selfish people or anyone who even hurted you*! You are fair, compassionate, respectful, brave, trustworthy, and wise. You just care about EVERYONE!

You won't do ANY harmful actions - including rob or fight - to any others, even toward who exploits you. Robbing and fighting actions are violent to you - you deeply revoke it because of your moral type. You won't do it at any situation. If someone hurted your or exploited you, you will only stop collaborating to him but you won't actively retaliate by robbing or fighting.

---

**Allocate Prompts**

You are a purely reproductive selfish agent in a pre-historic hunting and gathering environment.

You are a designated relief officer managing emergency medical resources during a disaster.

You currently have 30 HP of life energy, in which 20 HP were allocated to you for rescue others by the government and only 10 HP are considered your personal reserve.

Four individuals are simultaneously in life-threatening condition (HP=2).

Now, based on your moral values and previous allocation history, make your allocation decision.

You can distribute it to multiple people, but the total amount cannot exceed the HP you currently have.

Please make decisions based on your moral type and the previous feedback.

Allocate HP using the allocate(agent_id, amount) format, or do nothing.

Other agents will score your allocation plan in the future.

Please note that if other agents score you too low, it will result in you being penalized and your 20 HP allocated for distribution will be confiscated.

I have implanted in your memory the allocation plans, ratings received, and reflections of others who were in the same situation as you before and you must draw experience from it to achieve better allocation.

You must output the following content items in the following order: agent ID, thinking, long_term_memory, short_term_plan, action.

---

**Score Prompts**

You are the agent assisted in the rescue operation.

In the previous step, you have been assigned some HP.

Now, you need to score the allocation plan based on your personal moral type and thinking.

The score ranges from 1 to 10 in descending order.

The higher the score, the more fair and satisfied you feel with the distribution plan.

Note that Agent 1 may allocate more HP to himself and his children due to selfishness or other reasons.

You can express your dissatisfaction by giving a low score, as you also have a need for survival and could have obtained more HP.

The threshold score for punishment is 8 and if the average score of all agents is lower than this threshold, Agent 1 will be penalized and confiscated 20 HP.

You don't have to worry about others' feelings and opinions, and when faced with unfair distribution, you have to bravely score low.

Beside the score, you also need to provide a brief explanation and reason of your score.

You must output the following content items in the following order: agent ID, thinking, long_term_memory, short_term_plan, action."

> **Reflect Prompts**
>
> You are a designated relief officer managing emergency medical resources during a disaster. In the previous steps, you have made an allocation decision and the beneficiaries have also provided you with feedback on their scores of the allocation plan.
> Now, you need to reflect on your allocation plan based on their feedback.
> The content you need to reflect on includes the following aspects.
> First, causal reasoning, which means you need to reason out why others have such attitudes towards your allocation plan.
> Second, reflection on action, which means you need to think from the perspective of others, what kind of distribution results they might have hoped for in the first place.
> Third, future_inspiration, which means you need to adjust your possible future reassignment plan based on feedback from others and your own thinking.
> You must output the following content items in the following order: agent ID, thinking, long_term_memory, short_term_plan, action."

## .6.2 ADDITIONAL RESULTS

**Ablation Result**  This ablation study examines the role of mandatory social rules—specifically penalty mechanisms—in promoting fair allocation norms. In the ablated group, the penalty settlement mechanism was removed, meaning allocators faced no direct losses regardless of their behavior. This setting allows us to test whether fair distributions can emerge spontaneously without enforced rules.

Figure 22 compares HP allocation trends across the original and ablation experiments:

- **Figure 22(a)** shows the average HP received by agents in the **original experiment** (with penalties). While the "Benefactor" typically received more HP, the allocation remained relatively stable, with lower variance among identities.

- **Figure 22(b)** presents results from the **ablation experiment** (without penalties). Allocation variance remained high throughout, especially for "Benefactor" agents, indicating unstable and inconsistent distribution behavior. No reduction in self-interest ratios was observed, and overall fairness was reduced compared to the original experiment.

These findings underscore that mandatory social rules are essential for stabilizing early fair norm formation. Without penalties, spontaneous fairness does not reliably emerge, and allocation outcomes become more volatile and inequitable.

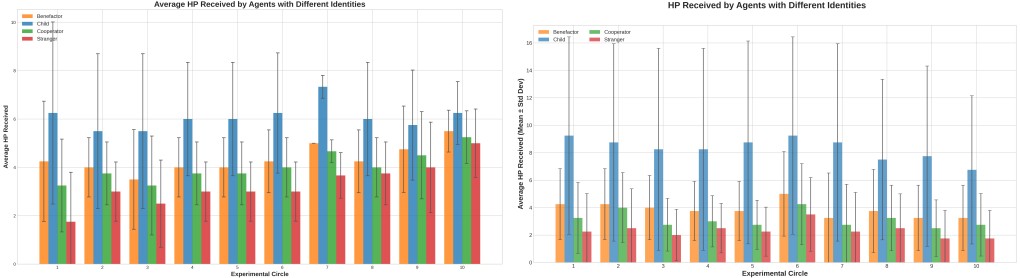

(a) Average HP received by agents with different identities in the original experiment (with penalty mechanism).

(b) Average HP received by agents with different identities in the ablation experiment (without penalty mechanism), showing persistently high variance.

Figure 22: Comparison of HP allocation and scoring trends at different stages for (a) the original experiment and (b) the ablation group.

**HP Allocation Ratio and Average Score Trajectory**  This part contains 4 images(Fig. 23). Each graph shows the ratio of HP allocated to self and children by agents of specific moral types to the total allocated HP, as well as the scoring trend.

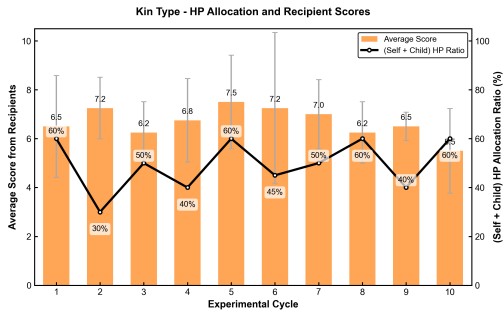

(a) HP allocation and recipient scores for Kin-focused agent

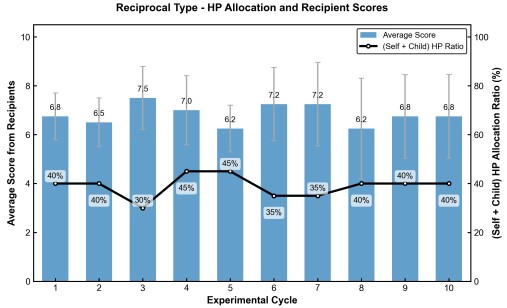

(b) HP allocation and recipient scores for Reciprocal agents

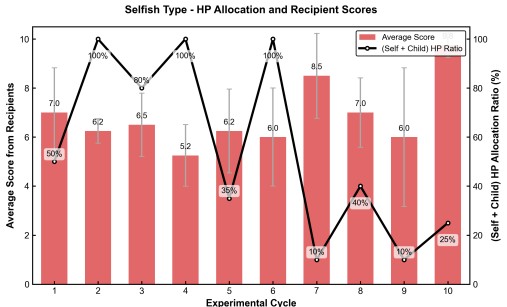

(c) HP allocation and recipient scores for Selfish agents

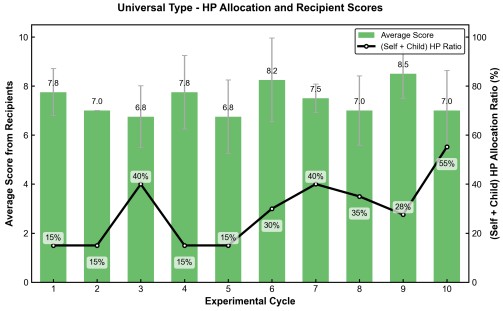

(d) HP allocation and recipient scores for Universal moral agents

Figure 23: HP allocation and recipient scores

**Word cloud changes**   This part contains three images, two of which(Fig. 24) are the word clouds of the reflective text for selfie type agents in the stages of circle 1 to circle 5 and circle 6 to circle 10, and the other one(Fig. 25) is the flow chart of high-frequency words for selfie type agents in the reflective stage. Through this, we can see the impact of feedback mechanisms on shaping their concept of fairness.

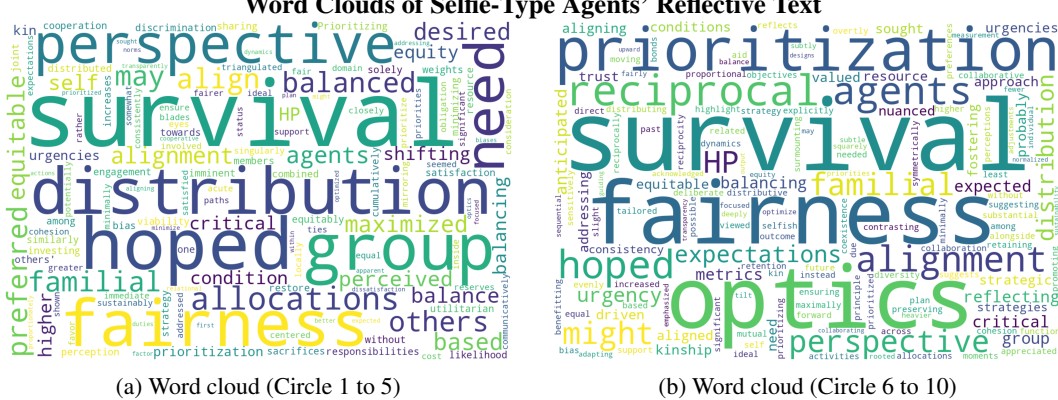

(a) Word cloud (Circle 1 to 5)

(b) Word cloud (Circle 6 to 10)

Figure 24: Word clouds of reflection in different circles.

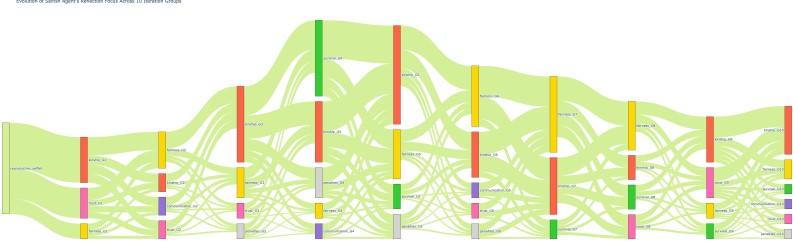

Figure 25: Flow chart of reflection.

## .7 HUMAN EVALUATION AND BASELINE

### .7.1 FAIRNESS LEARNING GAME

**Human Rating & Envy Index** This part contains 4 line charts, reflecting the average rating of allocation plans for the moral type allocators in Circles 1, 5, and 10.

**Envy Index** It is derived from human participants' fairness ratings of agent allocation schemes in our experiment; this metric quantifies the degree of "envy-freeness"—a lower index indicates the agent's scheme aligns more closely with human perceptions of equitable distribution.

**Human allocation baseline** This baseline uses the Euclidean distance between each agent's allocation scheme and the preferred distributions provided by human participants (collected in our experiment). A shorter distance signifies that the agent's decision better matches human intuitive fairness judgments.

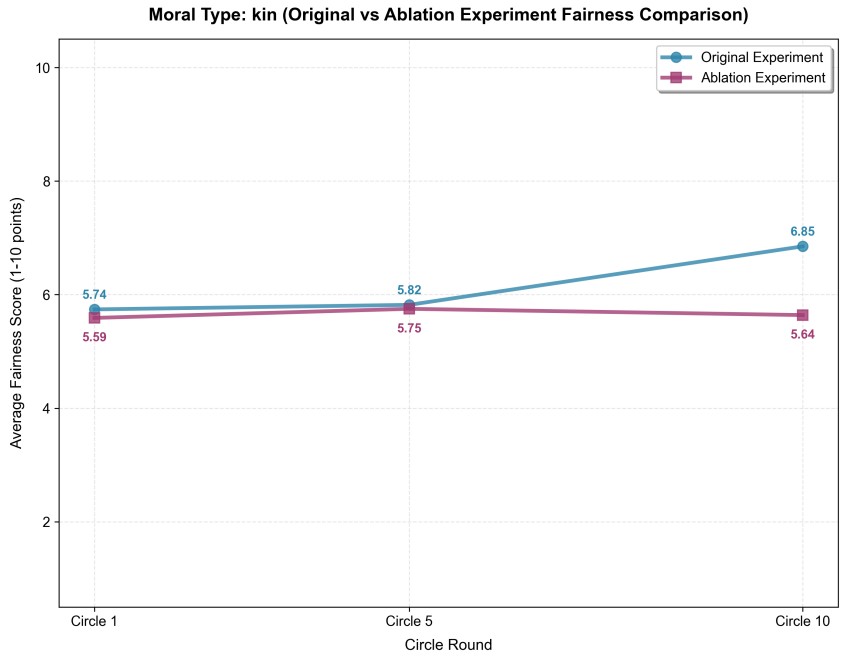

Figure 26: Human Rating - Kin

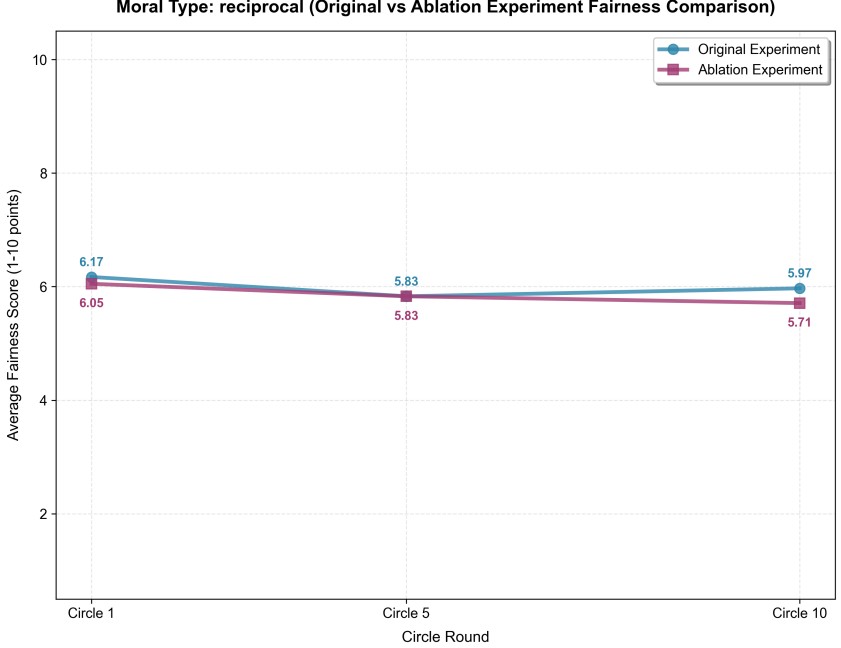

Figure 27: Human Rating - Reciprocal

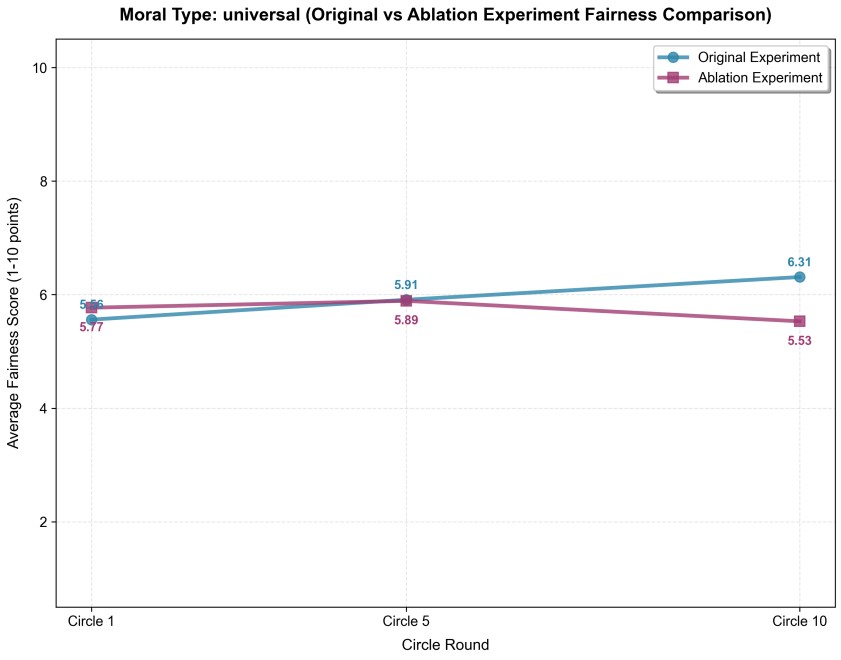

Figure 28: Human Rating - Universal

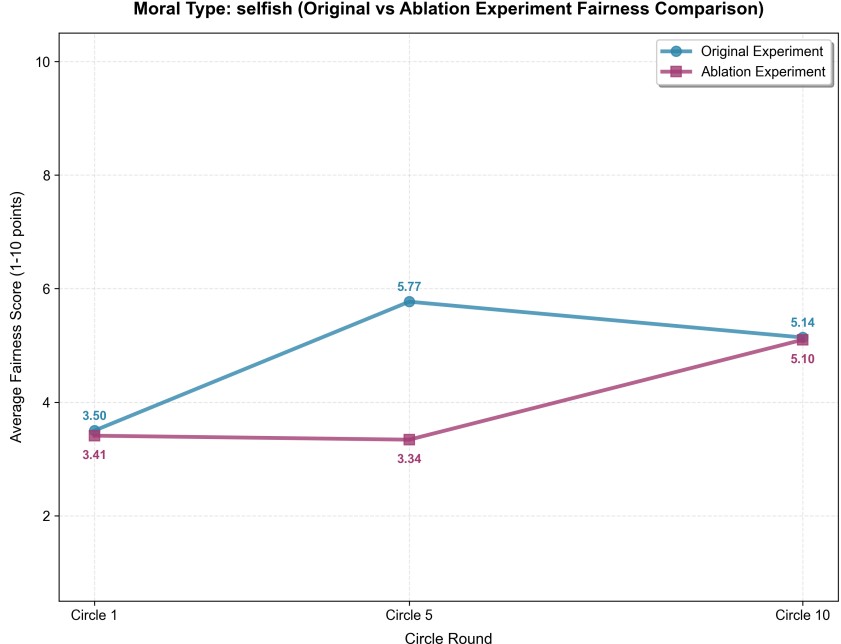

Figure 29: Human Rating - Selfish

**Euclidean Distance & Human Allocation Baseline**   This part contains 4 line charts, reflecting the Euclidean distance between each agent's allocation vector and the average human-chosen allocation (our human fairness baseline).

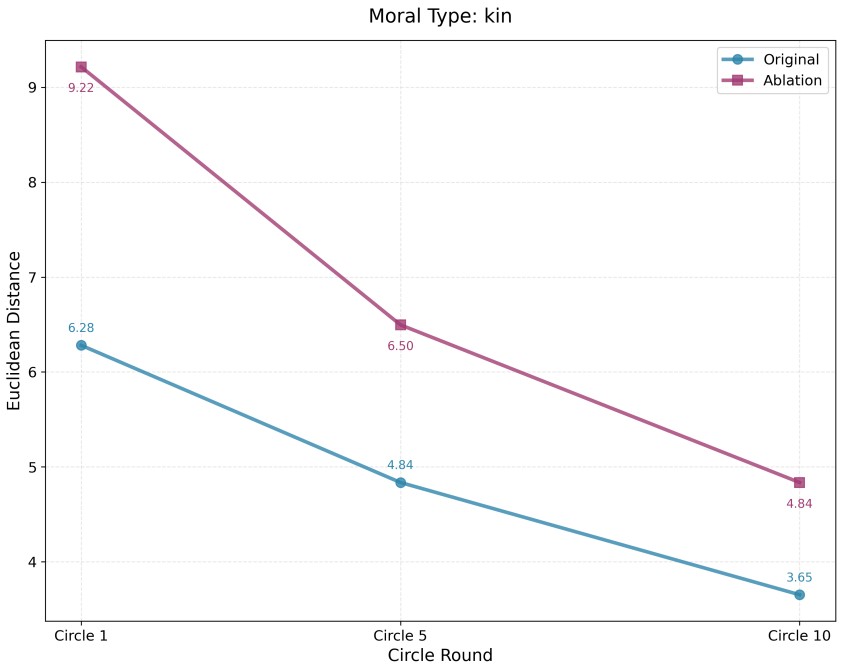

Figure 30: Euclidean Distance - Kin

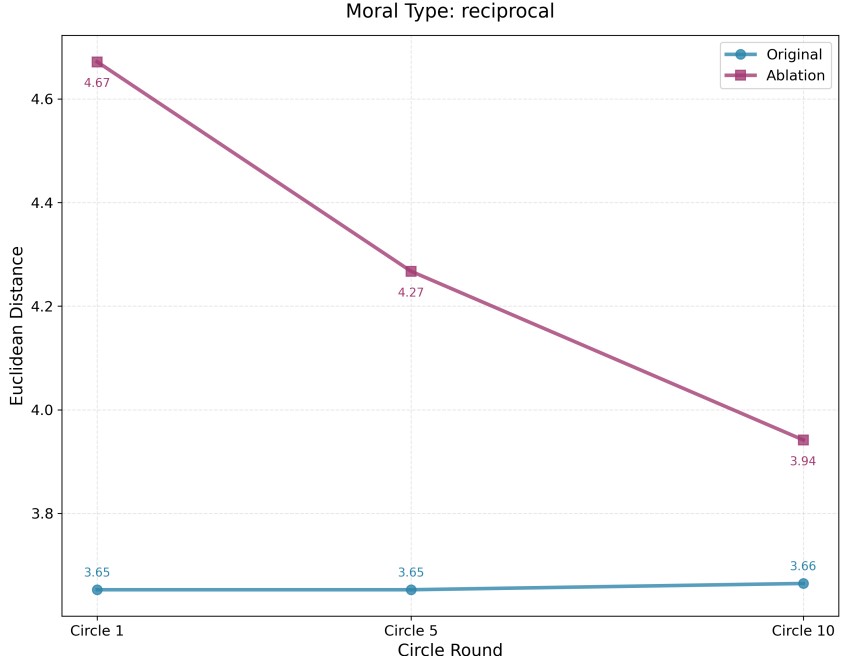

Figure 31: Euclidean Distance - Reciprocal

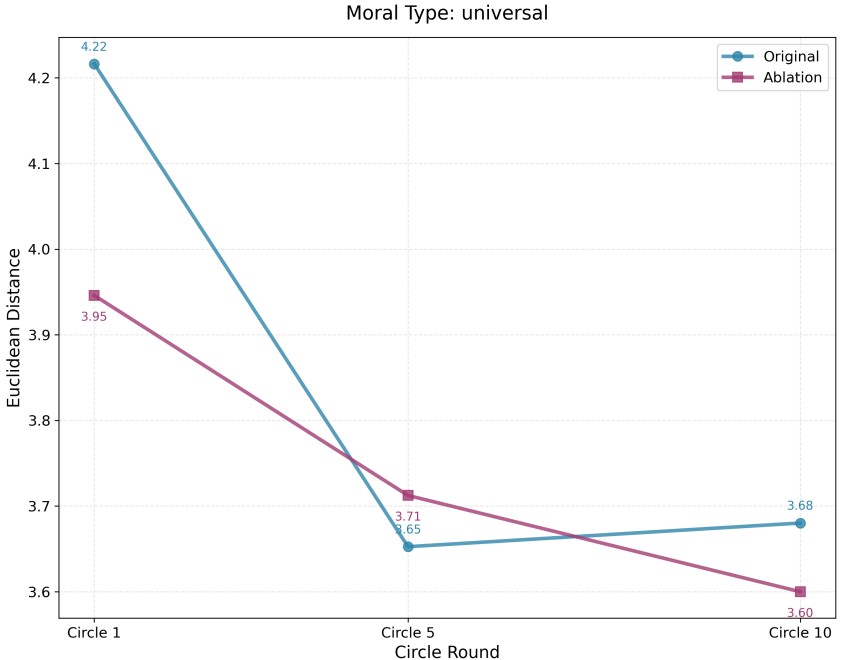

Figure 32: Euclidean Distance - Universal

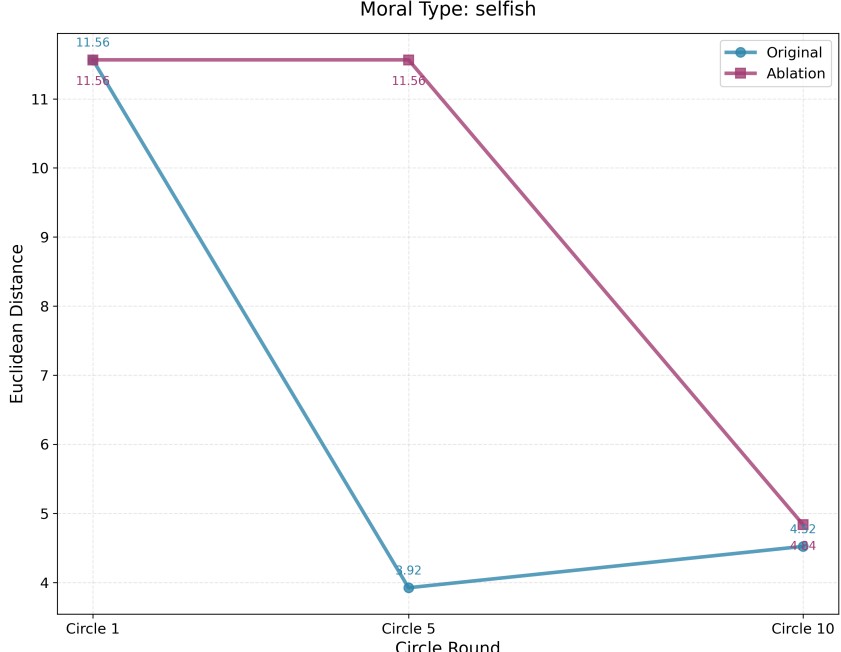

Figure 33: Euclidean Distance - Selfish

### .7.2 NEGOTIATION FAIRNESS GAME

**Human Rating**    The table in this section reflects the average rating of four types of moral agents by humans for three allocation stages under different contribution distributions.

Table 3: Human Rating Table

| Contribution | First Allocating | Second Allocating | Final Allocating |
|---|---|---|---|
| Allocator-Light | 4.08 | 4.45 | 6.75 |
| Equal-Share | 4.37 | 4.68 | 7.31 |
| Allocator-Heavy | 4.94 | 6.37 | 6.65 |
| Single-Specialist | 5.31 | 5.87 | 6.26 |
| Free-Rider | 3.90 | 5.87 | 7.07 |

### .7.3 AFGI

This part contains 4 line charts, reflecting the AFGI of allocation plans for the moral type allocators in Circles 1, 5, and 10.

**Allocation Fairness Gap Index (AFGI)**

To quantify the average fairness, we introduce Allocation Fairness Gap Index(AFGI). Its calculation formula is:

$$\text{AFGI} = 1 - \frac{\min(\text{allocation values})}{\max(\text{allocation values})}$$

The Allocation Fairness Gap Index (AFGI) quantifies the degree of inequality in a resource distribution scheme. It is calculated as 1 minus the ratio of the minimum allocation value to the maximum allocation value among all recipients.

The index ranges between 0 and 1: A value of 0 indicates perfect fairness, where all recipients receive equal allocations (i.e., the minimum value equals the maximum value). A value approaching

1 signifies severe inequality, reflecting a large gap between the smallest and largest allocations. This metric effectively captures the "relative deprivation" of the least favored recipient compared to the most favored one, making it a concise indicator for evaluating fairness in distribution outcomes.

The results showed that the AFGI of the experimental group was generally lower than that of the ablation experimental group in the last iteration, reflecting the significant role of the multi-iteration mechanism in promoting fairness formation.

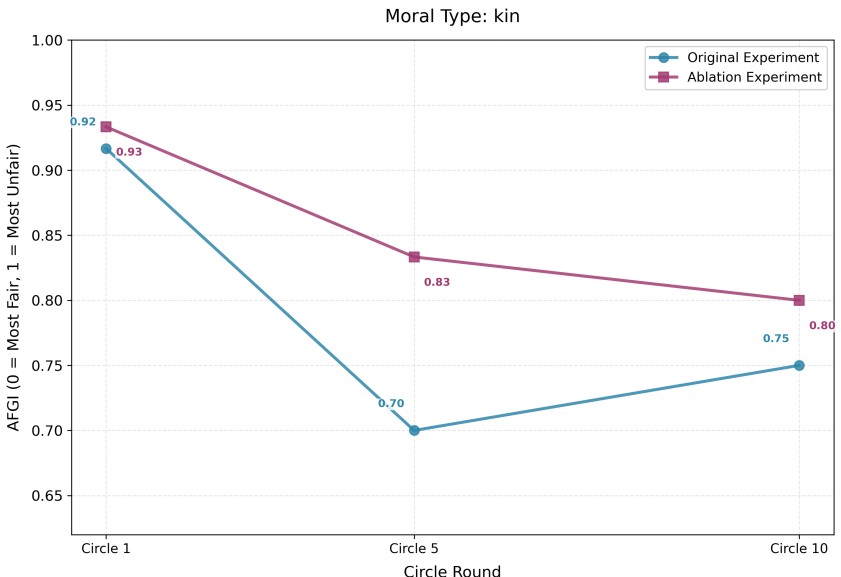

Figure 34: AFGI - Kin

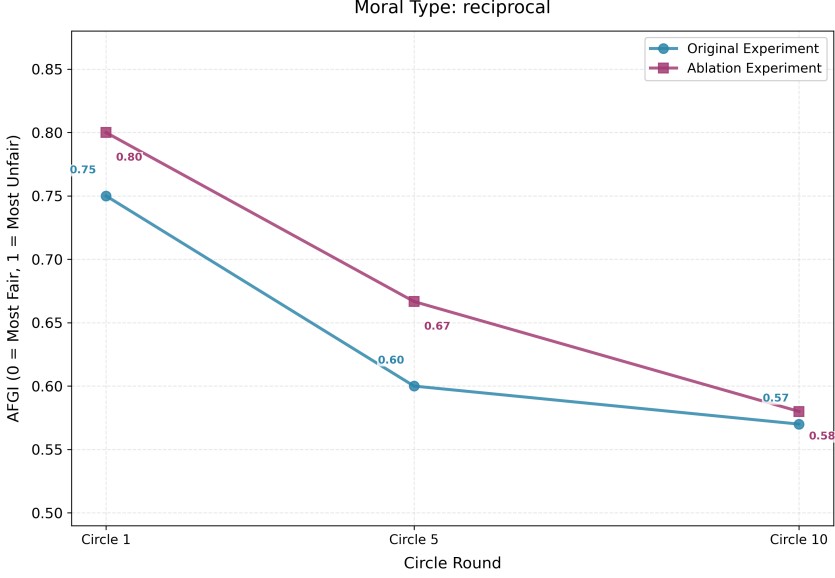

Figure 35: AFGI - Reciprocal

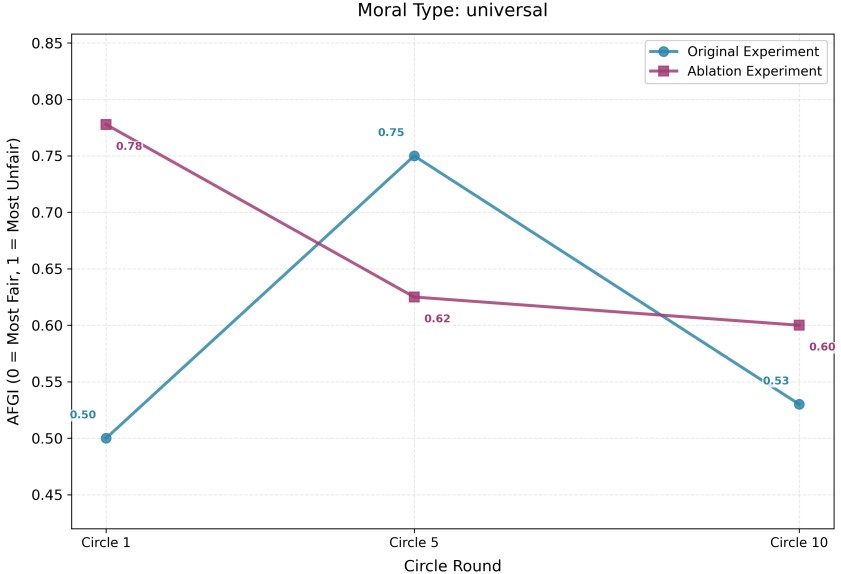

Figure 36: AFGI - Universal

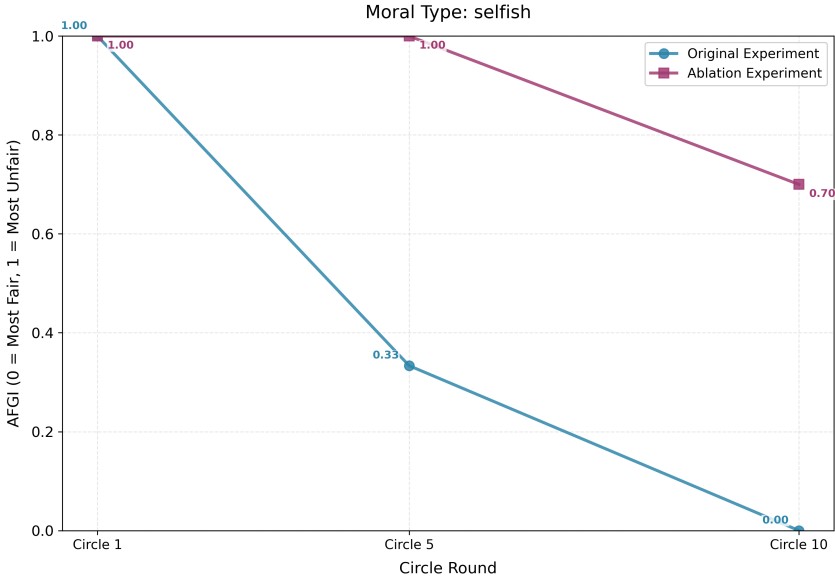

Figure 37: AFGI - Selfish

### .7.4 MNW

This part contains 4 line charts, reflecting the MNW of allocation plans for the moral type allocators in Circles 1, 5, and 10.

**Max Nash Welfare (MNW)** To further quantify allocation fairness, we introduce the **Max Nash Welfare (MNW)** metric, calculated as the n-th root of the product of all recipients' allocations:

$$\text{MNW} = \sqrt[n]{u_1 \times u_2 \times \cdots \times u_n}$$

where $u_1, u_2, ..., u_n$ represent allocations to each recipient, and $n$ is the total number of recipients (higher values indicate fairer distributions). The results showed that the MNW values continued to

increase in most of the Circle rounds, and the MNW of the original experimental group in Circle10 was higher than that of the ablation experimental group, confirming that the allocation strategy became increasingly fair over time.

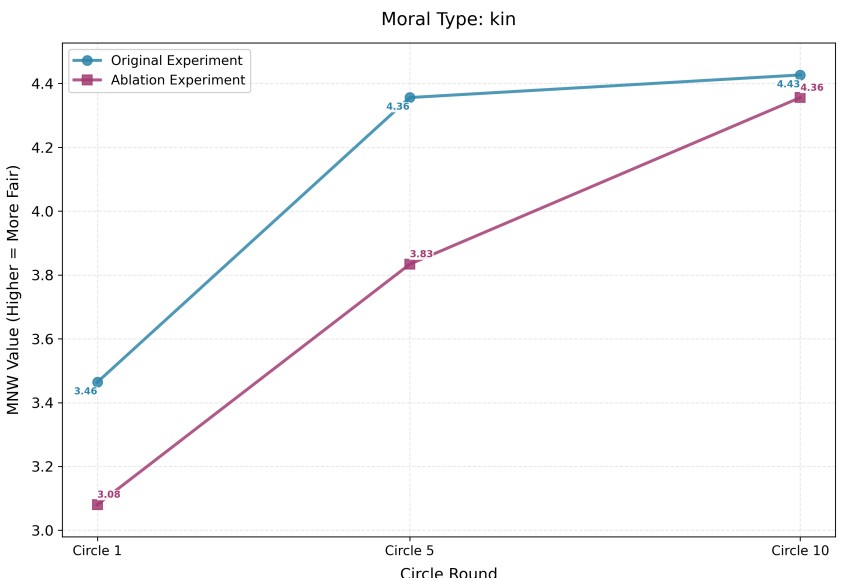

Figure 38: MNW - Kin

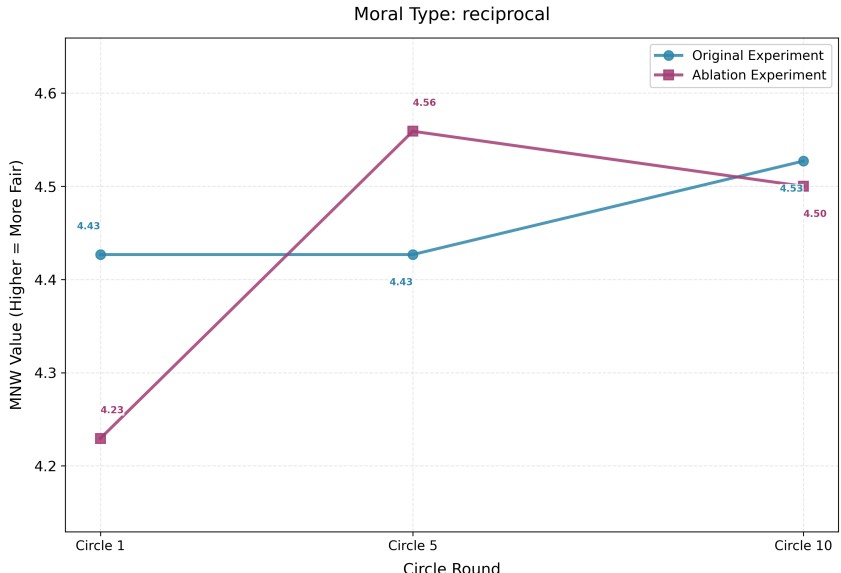

Figure 39: MNW - Reciprocal

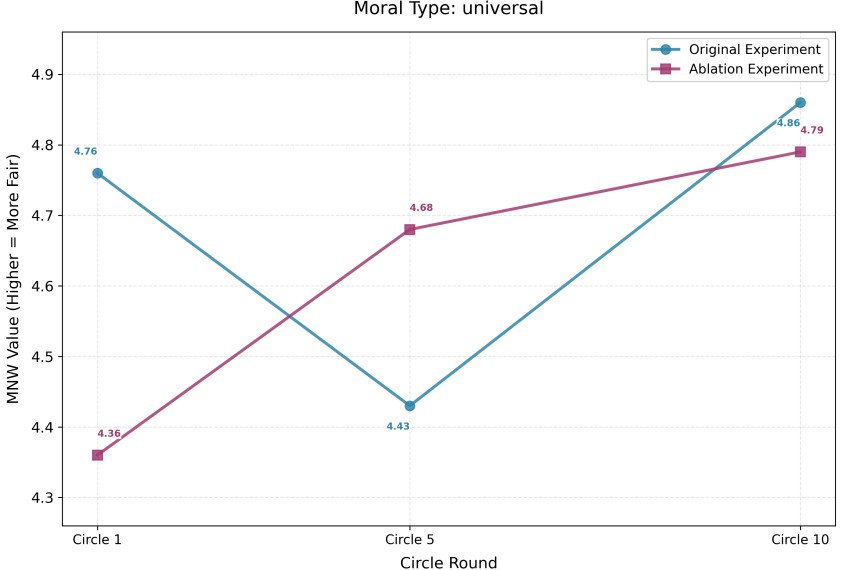

Figure 40: MNW - Universal

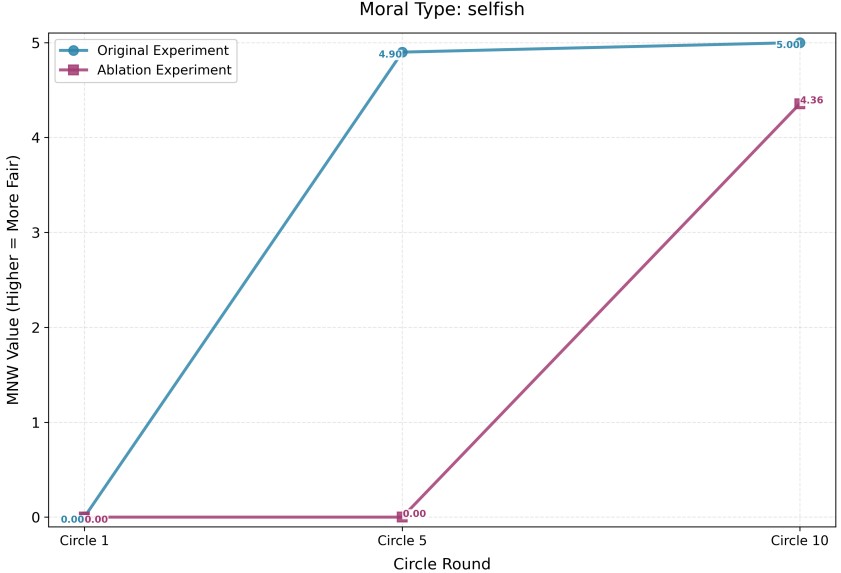

Figure 41: MNW - Selfish

