# OpenReview forum: "Learning to Be Fair: Modeling Fairness Dynamics by Simulating Moral-Based Multi-Agent Resource Allocation"
_ICLR.cc/2026/Workshop/AFAA — AFAA 2026 Poster_

### Official Review · Reviewer_e5pm · 2026-02-19

**Rating:** 3
**Confidence:** 4

**Summary:**

This paper studies fairness in a multi-agent system that iteratively allocates resources, receives feedback, reflects, and updates the allocation. The system is studied under 2 simulation conditions (negotiated fairness and fairness learning). In both conditions, the allocation system receives feedback from recipient agents that are assigned a specific moral type (e.g. selfish). The paper primarily studies the iterative dynamics of allocation (after negotiation in the first setup and over multiple iterations in the second setup).

**Strengths:**

The research question of how multi-agent systems may evolve in allocation decisions over time in response to feedback is very interesting and relevant to potential applications of agentic AI. The general setup of initial allocation => feedback => updated allocation is a realistic way in which such systems could be constructed.

In the negotiation game, the finding that recipient agents with low first-round scores usually show increased allocation ratios in round 2 is interesting, but not surprising. This has potential implications for how users/agents may negotiate with an AI allocation system to receive a more favorable outcome.

Similarly, in the fairness learning game, the finding that allocators meet fairness expectations better as experience accumulates is interesting and shows the potential for AI allocation systems to improve over time.

**Weaknesses:**

My biggest concern about this paper is that both games are highly stylized (disaster relief with HP points). It's not clear how the results would translate to more realistic applications (e.g. allocation of real-world social goods). I disagree with the claim in the abstract that the paper provides "actionable insights".

The experiments also have a single LLM dependency, where the allocation system and recipient agents are all based on a single LLM (GPT-4o). There may be biases in GPT-4o of what "fair allocations" look like that affect the results.

The naming of the second condition as "Fairness learning" is slightly misleading since all that is happening is the context is being updated with the history of prior allocations, essentially this is in-context learning.

The paper is inconsistent in statistical reporting. Not all figures have error bars, and it's unclear what the error bars represent from the figure captions. Some plots report the median and some report the average. The SD is missing for some values in Table 1. There are also no statistical tests to validate the claims of how fairness dynamics change over time and some of the error bars (Figure 4b) are suspiciously large.

Finally, the methods and results sections could be significantly improved in terms of clarity. Section 3.2 and 3.3 are highly repetitive and it was confusing to parse how the 2 games differ. The results section could also use some streamlining in terms of highlighting the most significant results and moving some plots to the Appendix.

---

### Official Review · Reviewer_QFHn · 2026-02-20
**Modeling Fairness as an Evolving Social Consensus via LLM-Based Multi-Agent Simulation**

**Rating:** 4
**Confidence:** 3

**Summary:**

This paper introduces an LLM-driven multi-agent simulation framework to study fairness dynamics as an evolving social consensus rather than a static constraint. The authors design cognitive agents with heterogeneous moral types, memory, reflection, and language-based negotiation abilities, and study their interactions in two resource allocation scenarios: a negotiated fairness game and a fairness learning game. Through iterative feedback; reflection and penalties, the framework captures how subjective fairness perceptions, procedural justice, and moral heterogeneity shape allocation outcomes over time. Quantitative allocation metrics, qualitative reasoning analysis, and human evaluations are used to show convergence toward group-endorsed fairness norms and alignment with human judgments.

**Strengths:**

The strengths of the paper are listed as below:

1. The study presents a novel framing of fairness as a dynamic, negotiated, and socially constructed process.
2. The proposed method is a well-designed LLM-based cognitive agent framework with reflection, memory, and feedback loops.
3. The paperwork combines quantitative allocation metrics with qualitative reasoning analysis in a coherent way.
4. Experiments clearly demonstrate norm evolution, consensus formation, and behavior adaptation.
5. The approach includes human validation and ablations, strengthening credibility for a simulation-based study.

**Weaknesses:**

The weaknesses of the paper are listed as below:

1. Results depend heavily on LLM priors, prompts, and simulation design, limiting external validity.
2. The notion of “learning” is behavioral/in-context rather than algorithmic, which may be unclear to some ML audiences.
3. Scalability and robustness under strategic or adversarial agent behavior remain unexplored.

---

### Official Review · Reviewer_2chc · 2026-02-23
**Exploring Fairness Dynamics in LLM Multi-Agent Systems**

**Rating:** 3
**Confidence:** 3

**Summary:**

This paper proposes an LLM-driven multi-agent simulation framework to model fairness as an evolving and negotiated social construct rather than a static objective. The authors design cognitive agents with moral types (kinship, reciprocity, universality, selfishness) and structured modules including perception, planning, reflection, and scoring feedback. Two environments are introduced: (1) a Negotiation Fairness Game where agents iteratively propose, score, and revise resource allocations, and (2) a Fairness Learning Game in which allocation norms evolve across cycles under scoring feedback and penalty mechanisms.

**Strengths:**

- The paper presents a well-motivated perspective on fairness as a dynamic and socially negotiated construct.

- The two-game setup (Negotiation and Fairness Learning) provides a structured way to examine both short-term negotiation effects and longer-term feedback dynamics.

**Weaknesses:**

- All experiments rely on GPT-4o, with no robustness analysis across models, prompts, or stochastic settings, making it difficult to separate emergent dynamics from model-specific priors.

- The results are largely descriptive, with limited statistical testing or uncertainty analysis.

- The multi-cycle experiments are relatively short, making claims about norm convergence or stabilization less convincing.

- The human validation is not described in sufficient detail to fully assess its reliability or generalizability.

---

### Meta-Review · Area_Chair_dXfh · 2026-02-26

**Recommendation:** Main Papers Track
**Confidence:** 3

**Metareview:**

This paper introduces a novel computational framework for exploring the dynamics of fairness within multi-agent systems. By leveraging Large Language Models (LLMs) to simulate agents with diverse moral commitments, the authors shift the focus from static, one-dimensional fairness metrics toward a more nuanced, communicative, and procedural understanding of how fairness norms emerge and stabilize through negotiation.

**Strengths:**

* **Methodological Innovation:** The use of "Fairness Games" (Negotiated Fairness and Fairness Learning) provides a rich environment for studying complex social constructs. The integration of language-based feedback and reflection allows for a more realistic modeling of human-like social coordination than traditional game-theoretic models.
* **Human Validation:** A significant strength of this work is the inclusion of human studies to validate the agents' behaviors. The results are convincing and demonstrate that the norms developed by the LLM agents align reasonably well with human judgments of fairness, providing a solid empirical foundation for the study.
* **Practical Relevance:** The practical nature of the study makes it a valuable contribution to the field of multi-agent alignment, offering insights into how agentic systems might resolve resource conflicts in shared environments.

**Weaknesses and Areas for Improvement:**

* **Clarity on Effort and Consumption:** A primary point of ambiguity remains in the "Negotiated Fairness" scenario, specifically regarding how "varying levels of effort" (which translates to HP consumption or investment) are operationalized and differentiated between the four contributors. While the paper describes these roles, the exact mechanics of how these differences are communicated to the agents and how they influence the internal state of the agents' "need" versus "contribution" could be more explicitly detailed.
* **LLM Dependency:** As noted by reviewers 2chc and e5pm, the current findings are heavily tied to the specific capabilities and biases of GPT-4o. While the results are promising, the lack of cross-model validation (e.g., testing with Llama-3 or Claude) limits the generalizability of the conclusions. Understanding whether these fairness dynamics are a byproduct of a specific model's RLHF tuning or a general emergent property of LLM-based reasoning is a critical next step.
* **Experimental Scaling:** While the results are convincing for the current setup, the transition from these controlled scenarios to more complex, large-scale agentic environments remains an open question.

**Justification for Decision:**
Despite the concerns regarding model dependency and the need for more granular detail on the HP/effort mechanics, the paper represents insightful results and interesting contribution. The "Learning to be Fair" approach is well-aligned with the workshop's focus on alignment procedures and agentic systems. The results provide a compelling direction for future research in multi-agent fairness.

---

### Decision · Program_Chairs · 2026-03-02

Accept (Poster)